# The microbiome interacts with the circadian clock and dietary composition to regulate metabolite cycling in the *Drosophila* gut

**Yueliang Zhang[1,2], Sara B Noya[1], Yongjun Li[1], Jichao Fang[2], Amita Sehgal[1]\***

[1]HHMI, Chronobiology and Sleep Institute, Perelman School of Medicine, University of Pennsylvania, Philadelphia, United States; [2]Institute of Plant Protection, Jiangsu Academy of Agricultural Sciences, Key Lab of Food Quality and Safety of Jiangsu Province, State Key Laboratory Breeding Base, Nanjing, China

## eLife Assessment

This study presents **valuable** findings about daily rhythm changes of the *Drosophila melanogaster* adult gut metabolome, which is shown to be dependent on the circadian clock genotype, dietary regime and composition, and gut microbiota. The phenomena observed are supported by **convincing** experimental evidence. The general descriptive approach limits the power of the proposed conclusions. The work will be of interest to a broad range of physiology specialists

*For correspondence:
amita@pennmedicine.upenn.edu

**Abstract** The gut microbiome plays a key role in the maintenance of host metabolic homeostasis and health. Most metabolic processes cycle with a 24-hour rhythm, but the extent to which the microbiome influences metabolite cycling under different conditions, such as variations in dietary composition, remains largely unknown. In this study, we utilized high temporal resolution metabolite profiling of the *Drosophila* gut to investigate the role of the microbiome in metabolite cycling. We find that the microbiome increases the number of oscillating metabolites despite the previous finding that it dampens transcript cycling in the gut. Time-restricted feeding also promotes metabolite cycling and does so to a larger extent in germ-free flies, thereby increasing cycling in these flies to levels comparable to those in microbiome-containing flies. Enhancement of cycling by the microbiome depends upon a circadian clock, which also maintains phase in the face of changes in the microbiome. Interestingly, a high protein diet increases microbiome-dependent metabolite cycling, while a high sugar diet suppresses it. Gene Ontology identifies amino acid metabolism as the metabolic pathway most affected by changes in the gut microbiome, the circadian clock, and timed feeding, suggesting that it is subject to regulation by multiple inputs. Collectively, our observations highlight a key role of the gut microbiome in host metabolite cycling and reveal a complex interaction with internal and external factors.

## Introduction

To adapt to the 24-hour light and temperature cycle caused by the Earth's rotation, organisms have evolved endogenous circadian clocks comprised of auto-regulating clock genes (*Sehgal, 2017*). Circadian clocks can be in the brain (i.e., central) or in peripheral tissues, and show differential responses to external entraining cues. In general, central clocks are mainly regulated by light signals, while dietary signals affect the rhythm of several peripheral circadian clocks (*Albrecht, 2012*; *Stephan, 2002*).

Central and peripheral clocks coordinate with each other to achieve synchronization of physiological and behavioral circadian rhythms. Disrupting circadian rhythms can result in a range of metabolic and neurological disorders (*Castanon-Cervantes et al., 2010*; *Rijo-Ferreira and Takahashi, 2019*; *Turek et al., 2005*), underscoring the importance of a well-synchronized circadian system.

The gut microbiome consists of microorganisms, including bacteria, viruses, protists, archaea, and fungi, that coexist symbiotically with hosts. Indeed, compelling evidence shows that gut microbiomes play a vital role in host nutrition, immunity, and neurodevelopment (*Flint, 2012*; *Seo et al., 2023*; *Thaiss et al., 2016b*). About 10–15% of the microbe species in the mammalian gut exhibit rhythms of abundance, and this is considered to be a key determinant of host health and fitness (*Liang et al., 2015*; *Thaiss et al., 2016a*; *Thaiss et al., 2014*). Time-restricted feeding (TF), which restricts food intake to a specific daily interval, enhances oscillations of the microbiome and improves health conditions in animals with obesity or metabolic syndrome (*Chaix et al., 2019*; *Lundell et al., 2020*; *Villanueva et al., 2019*). On the other hand, Western diets (high fat, high protein, and high sugar), which significantly increase the occurrence of metabolic diseases, can disrupt microbiome rhythms (*Hariri and Thibault, 2010*; *Ko et al., 2020*; *Na et al., 2013*). In addition, gut microbiomes regulate circadian rhythms of gene expression in metabolic tissues (*Thaiss et al., 2016a*; *Zhang et al., 2023*); however, their impact on host metabolite cycling remains largely unknown.

We previously reported that the gut microbiome in *Drosophila* does not cycle, but it regulates cycling of the host transcriptome. Specifically, it suppresses transcript rhythms in the gut while TF enhances these rhythms (*Zhang et al., 2023*). Here we sought to test whether the microbiome regulates metabolite oscillations and how the host clock and dietary manipulations interact with those cycles. Thus, we assayed metabolite rhythms in wild-type flies with and without a microbiome in different dietary paradigms and in clock mutant flies. Overall, we observed that the microbiome increases metabolite cycling. However, in the absence of a host clock, the microbiome decreases cycling of metabolites in total while dramatically affecting the phase of all. A TF feeding paradigm increases metabolite cycling, although less so in animals that contain a microbiome. On the other hand, a protein-rich diet enhances microbiome-dependent cycling, but this is not the case in sugar-rich diet. Overall, our results suggest the gut microbiome plays a critical role in maintaining cycles of metabolic activity in the gut, in a manner that depends on the host circadian clock, diet composition, and feeding behavior.

## Materials and methods
### Generation and maintenance of fly lines
The $w^{118}$ iso31, $per^{01}$ fly lines present in the study were maintained on standard cornmeal/molasses medium at 25°C under 12:12 LD conditions. The germ-free flies used in this study are the same as in the previous report (*Feltzin et al., 2019*; *Zhang et al., 2023*). The Newborn fly embryos within 12 hours were rinsed in 100% ethanol, dechorionated in 10% bleach for 2 minutes, then immediately rinsed three times in germ-free PBS. The germ-free embryos were transferred to autoclaved standard germ-free molasses-cornmeal-yeast medium containing 1 mM kanamycin, 650 μM ampicillin (61-238-RH, MediaTech), and 650 μM doxycycline (D9891, Sigma-Aldrich). Germ-free flies were maintained on germ-free mediums with three antibiotics for four generations and then maintained on the same medium without antibiotics for the next four generations. Bacterial contamination of flies was monitored by homogenizing larvae in germ-free PBS. Aspirate supernatant after brief centrifugation and monitoring the bacterial growth on DeMan, Rogosa, and Sharpe-agar plates (*Lekova, 2017*).

### Diets preparation and sample collection
Diets preparation followed the protocols below. Normal chow diets: 64.67 g of corn meal, 27.1 g of dry yeast, 8 g of agar, and 61.6 mL of molasses in 1 L ddH$_2$O. For high protein diets: 64.67 g of corn meal, 150 g of dry yeast, 8 g of agar, and 61.6 mL of molasses in 1 L of ddH$_2$O. For high sugar diets: 64.67 g of corn meal, 27.1 g of dry yeast, 8 g of agar, 61.6 mL of molasses, and 243 g of sucrose in 1 L of ddH$_2$O (*Bedont et al., 2021*; *Morris et al., 2012*). After autoclaving the diets at high temperature, temperature was allowed to drop to 40°C and then 10.17 mL of 20% Tegosept was added to each 1 L diet and mixed thoroughly. Germ-free conditions were maintained throughout. For all treatments, 5–7-day-old adults laid eggs on their respective diets (is031 and $per^{01}$ flies were fed normal chow

diets, while is031 flies were fed high protein or high sugar diets). The subsequent generation of newly hatched adults continued to feed on their respective diets for 3 days. Then, newly hatched female adult flies aged 5–7 days were separated by gender and subjected to the feeding paradigm in the presence of light-dark cycles beginning on day 5. In the timed feeding experiment, both ad libitum (AF) flies and those subjected to timed feeding (TF) were transferred to germ-free medium vials at Zeitgeber time 0 (ZT0), which corresponds to the actual time of 9:00 am. They were then relocated into either a new germ-free medium or a 1.1% agar vial at ZT10 (7:00 p.m.) for a 14-hour fast. In the ad lib feeding experiment, *per*[01] flies were fed normal chow diets, while iso31 flies received high protein and high sugar diets. AF flies were maintained in germ-free medium vials at all times, with germ-free fresh diets replaced according to the timed feeding schedule. Fly female guts were dissected on the fifth day at ZT0, ZT6, ZT12, and ZT18 within a 12:12 light-dark (LD) cycle. This dissection took place after the flies had been continuously fed for 4 days under ad lib or timed feeding conditions. For each condition, four biological repeat samples were collected both from microbiome-containing and germ-free female flies (>180 guts per sample, which corresponds to at least 10 mg for each sample). All experimental processes were carried out on ice, and samples quickly transferred to dry ice for storage after each sample was collected.

## Metabolite profiling

### Primary metabolites

Samples were extracted using the Matyash extraction buffer that includes MTBE, MeOH, and $H_2O$. The organic (upper) phase was dried down and submitted for resuspension and injection onto the LC while the aqueous (bottom) phase was dried down and submitted to derivatization for GC. 10 uL of methoxamine in pyridine was added to the aqueous phase, after which it was shaken at 30°C for 1.5 hours. Then 91 uL of MSTFA + FAMEs was added to each sample and shaking was continued at 37°C for 0.5 hours to finish derivatization. Samples were then vialed, capped, and injected onto the instrument. We use a 7890A GC coupled with a LECO TOF. 0.5 uL of derivatized sample is injected using a splitless method onto a RESTEK RTX-5SIL MS column with an Intergra-Guard at 275°C with a helium flow of 1 mL/min. The GC oven is set to hold at 50°C for 1 min then ramped at 20°C/min to 330°C and held for 5 min. The transferline is set to 280°C while the EI ion source is set to 250°C. The Mass spec parameters collect data from 85 m/z to 500 m/z at an acquisition rate of 17 spectra/s.

### Lipids

Samples were extracted using the Matyash extraction procedure, which includes MTBE, MeOH, and $H_2O$. The organic (upper) phase was dried down and submitted for resuspension and injection onto the LC while the aqueous (bottom) phase was dried down and submitted to derivatization for GC. It was resuspended with 110 uL of a solution of 9:1 methanol:toluene and 50 ng/mL CUDA. This was then shaken for 20 seconds, sonicated for 5 minutes at room temperature, and then centrifuged for 2 minutes at 16,100 rcf. The samples were then aliquoted into three parts. 33 uL were aliquoted into each of two vials with a 50 uL glass insert for positive and negative mode lipidomics. The last part was aliquoted into an Eppendorf tube to be used as a pool. The samples are then loaded up on an Agilent 1290 Infinity LC stack. The positive mode was run on an Agilent 6530 with a scan range of m/z 120–1200 Da with an acquisition speed of 2 spectra/second. Positive mode has between 0.5 and 2 uL injected onto an Acquity Premier BEH C18 1.7 µm, 2.1 × 50 mm column. The gradient consists of two mobile phases, of which only B is controlled by the software as follows: 0 min 15% (B), 0.75 min 30% (B), 0.98 min 48% (B), 4.00 min 82% (B), 4.13–4.50 min 99% (B), 4.58–5.50 min 15% (B) with a flow rate of 0.8 mL/min. The other sample aliquot was run in negative mode, which was run on Agilent 1290 Infinity LC stack, and injected on the same column, with the same gradient and using an Agilent 6546 QTOF mass spec. The acquisition rate was 2 spectra/second with a scan range of m/z 60–1200 Da. The mass resolution for the Agilent 6530 is 10,000 for ESI (+) and 30,000 for ESI (-) for the Agilent 6546.

### Biogenic amines

Sample extraction for biogenic amines is a liquid–liquid extraction using the Matyash extraction containing MTBE, methanol, and water and creating a biphasic partition. The polar phase is then dried down to completeness and run on a Waters Premier Acquity BEH Amide column. A short 4-minute

liquid chromatography method is used for separation of polar metabolites from a starting condition of 100% LCMS $H_2O$ with 10 mM ammonium formate and 0.125% formic acid to an end condition of 100% ACN: $H_2O$ 95:5 (v/v) with 10 mM ammonium formate and 0.125% formic acid. A Sciex Triple-ToF scans from 50 to 1500 m/z with MS/MS collection from 40 to 1000 selecting from the top 5 ions per cycle.

## Excreta quantification (Ex-Q) assay and feeding time course

For 24-hour feeding, we used the Ex-Q assay. Fifteen females were pre-fed on the three experimental diets described above for 3 days. On day 4, flies were transferred to the same food supplemented with 2.5 mg/mL of FD&C blue FCF brilliant blue dye for 48 hours. Flies were switched back to the specific diet without the dye for an extra 24 hours. Excrement was suspended into 150 uL of MilliQ water. Absorbance was measured at 620 nm. Raw data was divided by the number of flies and further normalized to the minimum value of the entire dataset. For time courses, females were pre-fed on the three experimental diets described above for 3 days. On day 4, groups of three flies were transferred to the same food supplemented with 2.5 mg/mL of FD&C blue FCF brilliant blue dye for 4 hours (i.e., for the ZT2 collection, flies were transferred to the blue food at ZT22 and collected at ZT2). Flies were collected and homogenized using a Tissue Lyser in 200 ul of PBS. After centrifugation to remove debris, absorbance was measured at 620 nm. Data was divided by the number of flies and further normalized to the minimum of each dataset, high protein and high sugar separately.

## Data processing and enrichment analysis

After acquisition, primary metabolomics data is converted from LECO peg files to NetCDF files for processing. Files are submitted to in-house data processor BinBase for annotation and a report is exported into an Excel file. In Excel, data is curated by blank reduction with a fold change of 3 or lower by using the max peak height from samples/average blank peak height. After reduction, peak heights are normalized by sum normalization. This is done using the un-normalized peak height of the feature divided by the mTIC (sum of the annotated metabolites) of the respective sample multiplied by the average mTIC of the samples. Normalized peak heights are then reported in an Excel document. Chromatograms first undergo a quality control check in which internal standards are examined for consistency of peak height and retention time. Raw data files are then processed using an updated version of MS-DIAL software, which identifies and aligns peaks and then annotates peaks using both an in-house mzRT library and MS/MS spectral matching with NIST/MoNA libraries. All MS/MS annotations are then manually curated by a lab member to ensure that only high-quality compound identifications are included in the final report. Enrichment analysis was conducted through the online software MetaboAnalyst 5.0.

## Statistical analysis

Statistical details of experiments can be found in the figure legends. Circadian statistical analysis was performed in R using JTK_CYCLEv3.1. JTK_p-value <0.05 and JTK BH.Q <0.2 deem as cycling metabolites (*Pang et al., 2022*).

# Results

## Microbiome enhances gut metabolite cycling

We showed previously that the *Drosophila* gut microbiome does not exhibit obvious oscillations in a 12:12 hours LD cycle, but it significantly reduces the number and strength of host transcript cycling (*Zhang et al., 2023*). To investigate the role of the microbiome in gut metabolite cycling, we assayed metabolite cycling at different times of day in microbiome-containing (AM) and germ-free (AS) fed ad libitum (*Figure 1A*). Metabolic profiling was performed separately to characterize primary metabolites, lipids, and biogenic amines (MetaboLights datasets MTBLS8819 contain the raw data). Principal component analysis (PCA) of the total expressed primary metabolites did not show significant separation between AM and AS flies at any time of day, indicating that this class of molecules is not changed by loss of the microbiome. However, a clear separation between the two groups was observed for lipids and biogenic amines (*Figure 1—figure supplement 1A–C*).

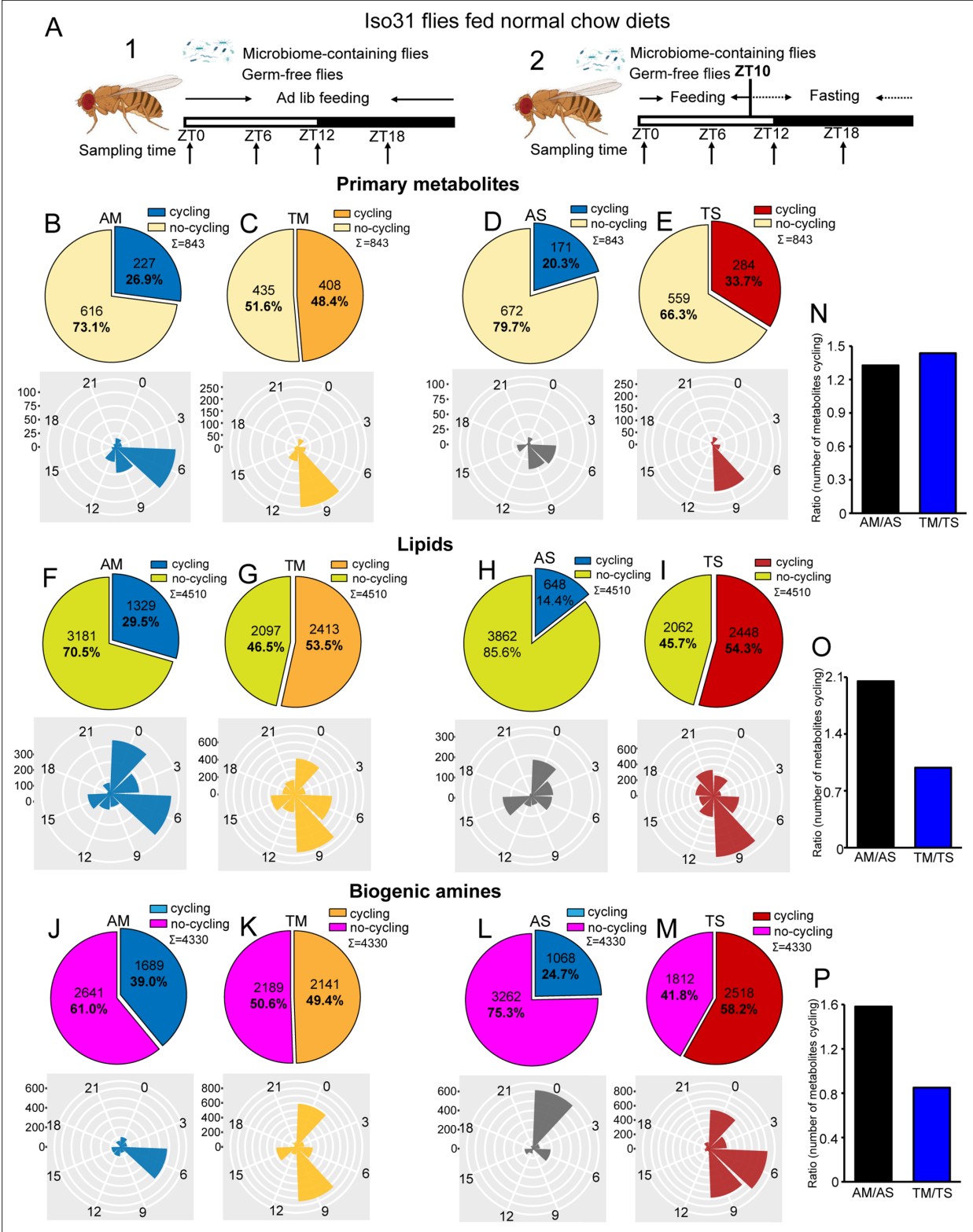

**Figure 1.** Microbiome reduces the effect of timed feeding on the cycling of lipids and biogenic amines. (**A**) Schematic showing the experimental protocol for (1) analysis of gut metabolite cycling in microbiome-containing (AM) and sterile (AS) iso31 flies fed on normal chow diets under ad lib and (2) timed feeding (TF) conditions, respectively. (**B–E**) Pie chart depicting the fraction of cycling primary metabolites and polar histogram plots of the peak phase for cycling primary metabolites in microbiome-containing and sterile flies under ad lib and timed feeding conditions (AM, TM, AS, and TS), respectively, using a JTK_cycle cutoff of p<0.05. (**F–I**) Pie chart depicting the fraction of cycling lipids and polar histogram plots of the peak phase

*Figure 1 continued on next page*

*Figure 1 continued*

for cycling lipids in AM, TM, AS, and TS flies, respectively, using a JTK_cycle cutoff of p<0.05.(**J–M**) Pie chart depicting the fraction of cycling biogenic amines and polar histogram plots of the peak phase for cycling biogenic amines in AM, TM, AS, and TS flies, respectively, using a JTK_cycle cutoff of p<0.05. (**N–P**) The ratio of primary metabolites, lipids and biogenic amines that cycle in microbiome-containing versus sterile flies under ad lib feeding (AM/AS) and timed feeding (TM/TS) conditions, respectively.

The online version of this article includes the following figure supplement(s) for figure 1:

**Figure supplement 1.** Timed feeding has less effect on cycling metabolites in microbiome-containing flies.

**Figure supplement 2.** The percentage of unique and shared cycling metabolites in different treatment groups.

**Figure supplement 3.** The amplitude of metabolite cycling is sometimes, but not always, increased by the presence of a microbiome or by timed feeding.

**Figure supplement 4.** Lipids and biogenic amines show higher abundance in microbiome-containing flies.

**Figure supplement 5.** Abundance analysis of all metabolites in microbiome-containing and sterile flies.

**Figure supplement 6.** Loss of the circadian clock or high protein or high sugar diet decreases metabolite cycling in microbiome-containing flies.

Given our transcriptomic results, we were expecting that the gut microbiome would reduce the number of cycling metabolites; however, we observed the opposite. JTK cycling analysis of the three different types of metabolites revealed that the likelihood of oscillations increased in AM flies compared to AS flies. 26.9% (227 of 843) primary metabolites were rhythmic in AM flies compared to 20.3% (171 of 843) in AS flies. A further reduction in AS was observed for lipids whose rhythms dropped from 29.5% (1329 of 4510) cycling in AM to 14.4% (648 of 4510) in AS. Similarly, the proportion of biogenic amines cycling decreased from 39.0% (1689 of 4330) to 24.7% (1068 of 4330) (JTK_p-value <0.05) (*Figure 1B, D, F, H, J and L*). In addition, only about half of the cycling metabolites in germ-free flies are shared with microbiome-containing flies (*Figure 1—figure supplement 2*). Consistent with the role of the microbiome in favoring metabolic rhythms, lipids and biogenic amines that cycle in both AM and AS flies have a higher amplitude in AM flies (*Figure 1—figure supplement 3A–C*). Although the microbiome did not significantly change the overall distribution of phases for the cycling primary metabolites and lipids, it had a substantial effect on the phase of biogenic amine oscillations that shifted from peaking in the middle of the day (ZT6) in the AM flies to a peak in the early morning (ZT0) in AS flies (*Figure 1B, D, F, H, J and L*). Morning peaks were also noted for lipids under almost all conditions, which may be driven by the morning peak of feeding.

Analysis of the abundance of cycling metabolites in AM, compared to their non-cycling counterparts in AS, also revealed that lipids and biogenic amines had higher expression levels in AM flies compared to AS flies. However, there was no significant difference between AM and AS flies in abundance of primary metabolites (*Figure 1—figure supplement 4A, E and I*). Likewise, the overall abundance, regardless of cycling, of lipids and biogenic amines was higher in AM flies but primary metabolites were equivalent in AS and AM (*Figure 1—figure supplement 5A–C*).

Overall, this indicates that the microbiome increases the cycling of all metabolites in the gut but has more dramatic effects on lipids and biogenic amines.

## Microbiome has less effect on metabolite cycling under timed feeding conditions

Timed feeding affects host metabolism and benefits host health, according to several studies (*Chaix et al., 2019*; *Gill et al., 2015*; *Lundell et al., 2020*). In this study, we aimed to determine how microbiome-mediated metabolite cycling interacts with a timed feeding paradigm in which flies are provided with food only from ZT0 to ZT10. Similarly to ad libitum conditions, PCA of the total expressed primary metabolites showed only minor separation between the four groups— AM, AS, microbiome-containing flies subjected to timed feeding (TM), and germ-free flies subjected to timed feeding (TS)—but a clear separation was observed for lipids and biogenic amines just as seen for AM and AS flies (*Figure 1—figure supplement 1A–C*). Consistent with the results of our previous study on transcript cycling, we found that timed feeding significantly increased the number of metabolites cycling in germ-free and microbiome-containing flies (*Figure 1B–M*, *Figure 1—figure supplements 2B,C,E,F, H, I and 6A–D*).

Time-restricted feeding coordinated the phase of all three-metabolite classes with TM and TS flies displaying very similar phase distributions. Interestingly, primary metabolites peaked around a

single time of day whilst lipids and biogenic amines displayed two distinct phases in timed feeding conditions (*Figure 1B–M*). Moreover, the observation that TM and TS flies display similar metabolite phases differs from what happens in ad lib conditions in which the phase of biogenic amines and lipids changes in the absence of a microbiome. In particular, metabolites peaking at ZT9 increased with TF in all metabolite groups. This could suggest an anticipation of sleep at the end of the feeding window. Interestingly, a smaller proportion of cyclers is specific for the ad libitum condition compared to the TF in both sterile and microbiome-containing flies (*Figure 1—figure supplement 2*).

Analysis of abundance of cycling metabolites under ad lib feeding and timed feeding conditions revealed that, in general, rhythmic metabolites tend to have higher overall expression levels compared to non-rhythmic metabolites, and this phenomenon is more prominent in the case of cycling biogenic amines (*Figure 1—figure supplement 4A–L*). Additionally, we also looked at the amplitude of metabolites that cycle in both microbiome-containing and microbiome-free flies under the two feeding paradigms. In this case, the three metabolite groups did not always show a larger amplitude in the timed fed groups, which slightly contradicts the potential benefits of this type of intervention (*Figure 1—figure supplement 3D–I*).

To further determine whether timed feeding modulates the effect of the microbiome, we compared the number of metabolites cycling with TF in AS and AM flies and found that loss of the microbiome had a larger effect in ad lib fed flies. This was reflected in the ratios depicting metabolite cycling in microbiome-containing versus germ-free flies, such that the AM/AS ratio was higher than the TM/TS for lipids and biogenic amines, although not for primary metabolites (*Figure 1N–P*). Moreover, the TS/AS ratio for lipids and biogenic amines was also higher than TM/AM (*Figure 1—figure supplement 1D–F*). These data suggest that loss of the microbiome has less of an effect on timed fed flies than on ad lib fed, and, conversely, timed feeding has less effect on microbiome-containing flies. Thus, time-restricted feeding improves cycling and may compensate for the lack of a microbiome.

## Effects of the microbiome on the cycling metabolome are modulated by the circadian clock

To determine whether circadian clocks play a role in the enhanced metabolite cycling in microbiome-containing flies, we next subjected $per^{01}$ flies with ($per^{01}$-AM) and without ($per^{01}$-AS) a microbiome to ad lib feeding conditions and examined the cycling of metabolites in their guts (*Figure 2A*). The results showed that the PCA for the three metabolite classes differentiated $per^{01}$-AM and $per^{01}$-AS flies (*Figure 2—figure supplement 1A–C*). Surprisingly, loss of the microbiome increased metabolite cycling in $per^{01}$ flies, not for primary metabolites, but increasing the proportion of lipids cycling from 17.8% to 22.2% and almost duplicating the proportion of biogenic amines cycling from 18% in $per^{01}$-AM to 30.5% cycling features in $per^{01}$-AS flies (*Figure 2B, D and F*). Moreover, all cycling metabolites exhibited completely opposite phases in $per^{01}$-AM relative to $per^{01}$-AS flies (*Figure 2C, E and G*). Phases also changed for cycling metabolites shared between AM and AS (*Figure 2—figure supplement 2A–C*). Along these lines, we found an even lower overlap of cycling metabolites between AM and AS in $per^{01}$ flies (*Figure 1—figure supplement 2*), indicating that the clock modulates not just the phase but also the identity of cyclers.

Analysis of the abundance of cycling metabolites in $per^{01}$ flies revealed that, in general, cycling lipids and biogenic amines have higher expression in AM compared with their non-cycling equivalents in AS, whereas primary metabolites show minor difference between the two groups (*Figure 2—figure supplement 3A–C*). This result is consistent with the comparison of abundance in microbiome-containing and germ-free iso31 flies.

These datasets suggest a complex interaction between the microbiome and the host clock. It was noticeable that lack of a host clock resulted in reduced metabolite rhythms in microbiome-containing flies (*Figure 1—figure supplements 2A, D, G and 6A, E*) but enhanced cycling in microbiome-free flies (*Figure 1—figure supplements 2J, K, L and 6B, F*). This also caused profound changes in phase (*Figure 2*). Overall, it appears that the host clock stabilizes metabolite rhythms against changes in the microbiome.

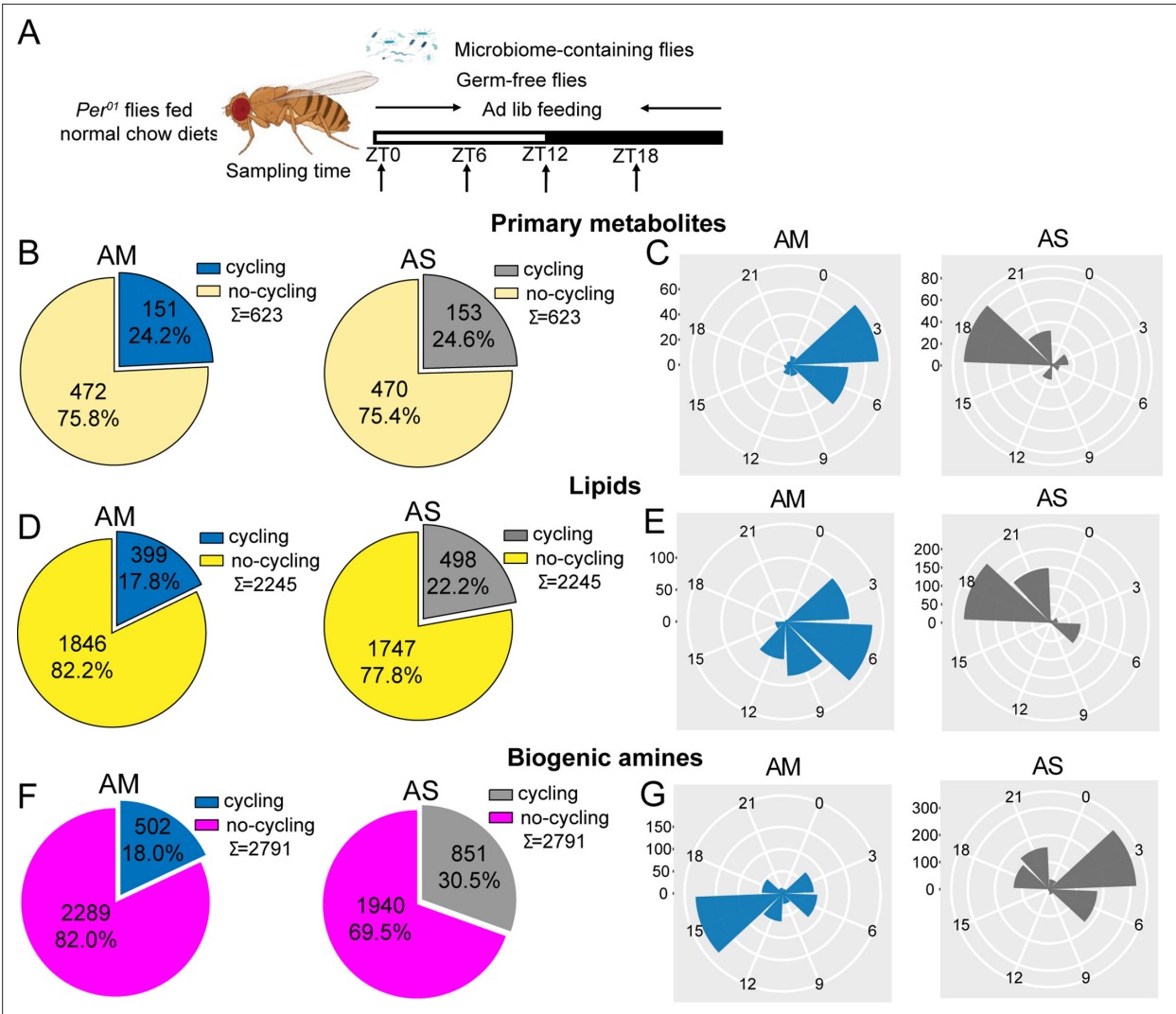

**Figure 2.** The effect of the microbiome on metabolite cycling depends on a functional circadian clock. (**A**) Schematic showing the experimental protocol for analysis of gut metabolite cycling in microbiome-containing (AM) and sterile (AS) *per01* flies fed on normal chow diets. (**B**) Pie chart depicting the fraction of cycling primary metabolites, and (**C**) polar histogram plots of the peak phase for cycling primary metabolites in AM and AS *per01* flies fed on normal chow diets, using a JTK_cycle cutoff of p<0.05. (**D**) Pie chart depicting the fraction of cycling lipids and (**E**) polar histogram plots of the peak phase for cycling lipids in AM and AS *per01* flies fed on normal chow diets, using a JTK_cycle cutoff of p<0.05. (**F**) Pie chart depicting the fraction of cycling biogenic amines and (**G**) polar histogram plots of the peak phase for cycling biogenic amines in AM and AS *per01* flies fed on normal chow diets, using a JTK_cycle cutoff of p<0.05.

The online version of this article includes the following figure supplement(s) for figure 2:

**Figure supplement 1.** Principal component analysis (PCA) of total expressed metabolites in *per01* flies fed on normal chow diets, iso31 flies fed on high protein and high sugar diets.

**Figure supplement 2.** Phase changes for cycling metabolites shared between *per01*-AM and *per01*-AS.

**Figure supplement 3.** Effect of the microbiome on expression levels of cycling metabolites in clock mutants and under different dietary conditions.

## Microbiome enhances metabolite cycling in the presence of high protein diets and decreases cycling with high sugar diets

Changes in diet composition can significantly affect gut microbiome rhythms (*Flint, 2012*; *Frazier et al., 2022*). Therefore, we also explored the cyclic metabolome of flies with or without a microbiome and fed on two different diets; a high sugar and a high protein diet (*Figures 3A and 4A*). From the PCA , we noticed that high sugar diets trigger metabolic differences between flies that contain a

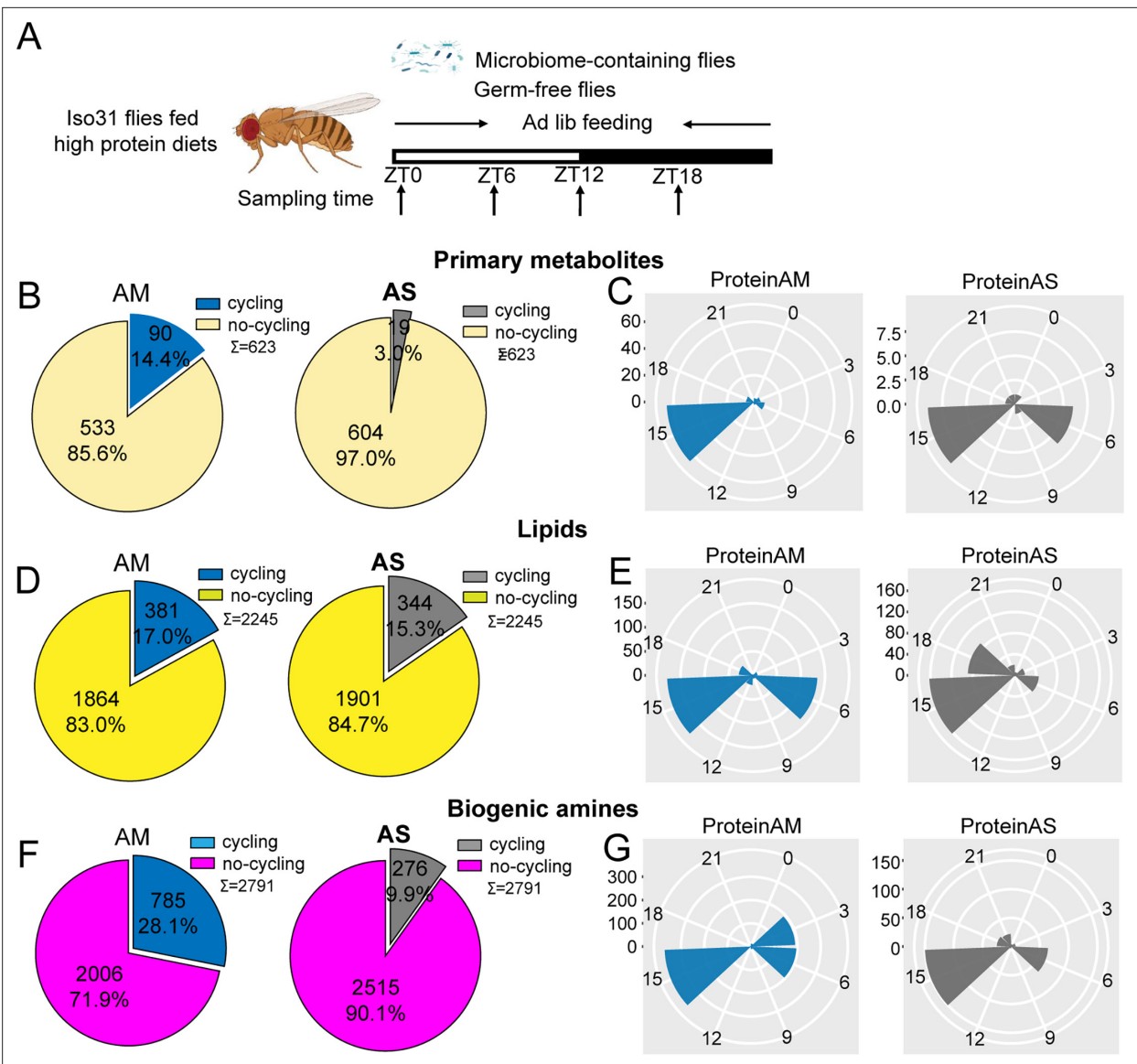

**Figure 3.** Microbiome increases the number of cycling of metabolites in flies fed on high protein diets. (**A**) Schematic showing the experimental protocol for analysis of gut metabolite cycling in microbiome-containing (AM) and sterile (AS) Iso31 flies fed on high protein diets. (**B**) Pie chart depicting the fraction of cycling primary metabolites, and (**C**) polar histogram plots of the peak phase for oscillating primary metabolites in AM and AS iso31 flies fed on high protein diets, using a JTK_cycle cutoff of p<0.05. (**D**) Pie chart depicting the fraction of cycling lipids, and (**E**) polar histogram plots of the peak phase for oscillating lipids in AM and AS iso31 flies fed on high protein diets, using a JTK_cycle cutoff of p<0.05. (**F**) Pie chart depicting the fraction of cycling biogenic amines, and (**G**) polar histogram plots of the peak phase for oscillating biogenic amines in AM and AS iso31 flies fed on high protein diets, using a JTK_cycle cutoff of p<0.05.

microbiome and microbiome-free flies; however, high protein diets do not prompt such difference (*Figure 2—figure supplement 1 D–F and G–I*).

On high protein diets, the lack of a microbiome reduced the number of rhythmic metabolites (*Figure 3B, D and F*). This was consistent with the role of the microbiome in promoting metabolic rhythms. However, the effects of a high sugar diet diverged substantially from those of a high protein diet. With the high sugar diet, rhythmic lipids and biogenic amines increased in germ-free flies, similar to what we observed in a clock mutant background (*Figure 4B, D and F*). Indeed, when the three metabolite groups were analyzed together, flies on a high sugar diet showed enhanced cycling of the metabolome when the microbiome was absent (*Figure 1—figure supplement 6I and J*).

We also looked at phases of cycling metabolites. In flies fed a high protein diet, a predominant peak around ZT15 was observed for all metabolites regardless of the presence or absence of a microbiome

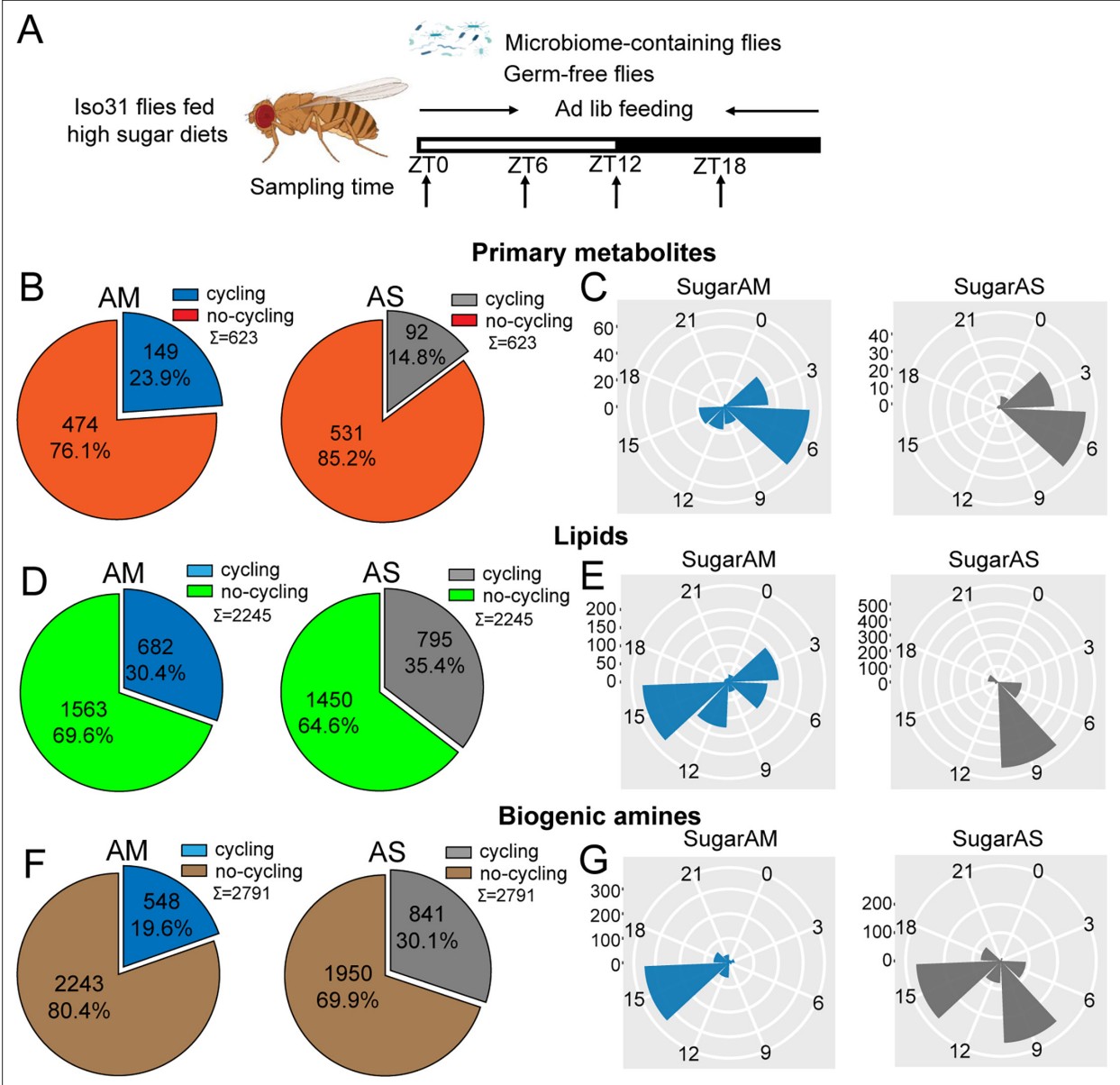

**Figure 4.** Microbiome decreases the number of cycling metabolites in iso31 flies fed on high sugar diets. (**A**) Schematic showing the experimental protocol for analysis of gut metabolite cycling in microbiome-containing (AM) and sterile (AS) Iso31 flies fed on high sugar diets. (**B**) Pie chart depicting the fraction of cycling primary metabolites, and (**C**) polar histogram plots of the peak phase for cycling primary metabolites in AM and AS iso31 flies fed on high sugar diets, using a JTK_cycle cutoff of p<0.05. (**D**) Pie chart depicting the fraction of cycling lipids, and (**E**) polar histogram plots of the peak phase for cycling lipids in AM and AS iso31 flies fed on high sugar diets, using a JTK_cycle cutoff of p<0.05. (**F**) Pie chart depicting the fraction of cycling biogenic amines and (**G**) polar histogram plots of the peak phase for cycling biogenic amines in AM and AS iso31 flies fed on high sugar diets, using a JTK_cycle cutoff of p<0.05.

The online version of this article includes the following figure supplement(s) for figure 4:

**Figure supplement 1.** Feeding rhythms do not show an obvious link with metabolite cycling in different genotypes and under different conditions.

(*Figure 3C, E and G*). This suggests that protein metabolism imposes a specific metabolic rhythm that is not dependent on the gut microbiome. In case of the high sugar diet, we noticed that the absence of microbes changed the phase distribution of both lipid and biogenic amines but not of primary metabolites (*Figure 4C, E and G*).

We also compared the dataset generated from flies fed a standard diet versus these nutrient-specific paradigms. For flies with a normal microbiome, the fraction of cycling metabolites decreased from 33.5% on standard diet to 22.2% and 24.5% with a high protein and high sugar diet, respectively

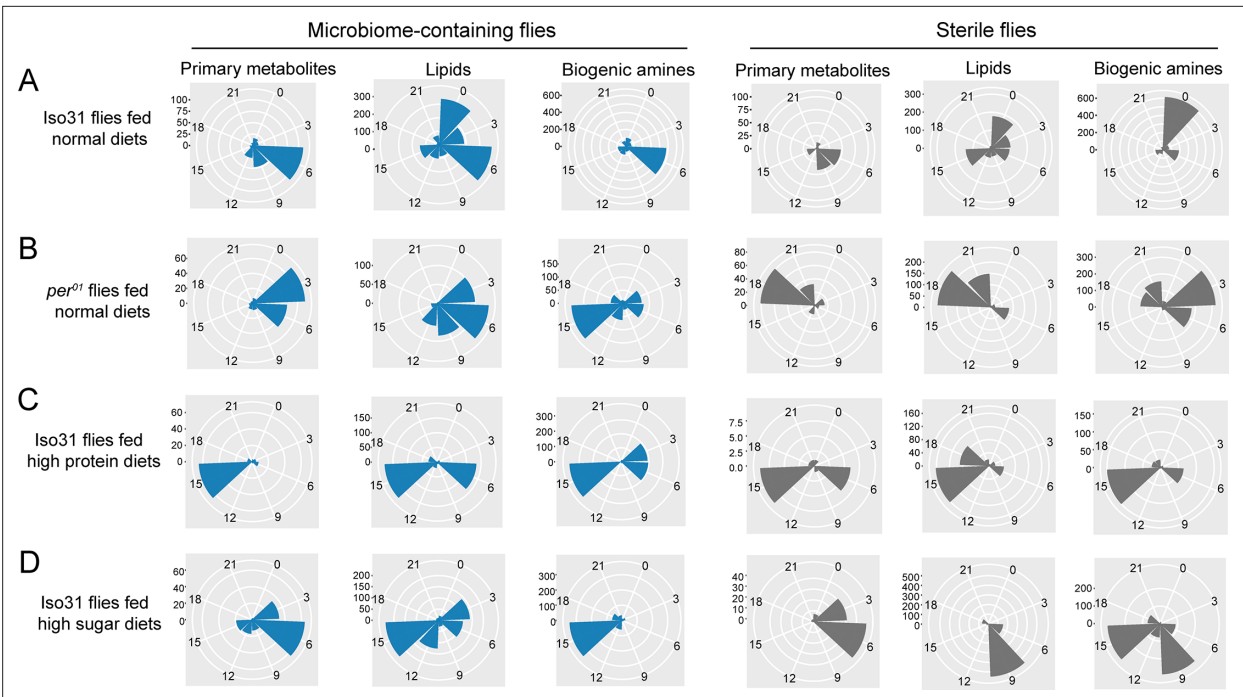

**Figure 5.** Effects of the circadian clock and dietary composition on the phase of metabolite cycling in microbiome-containing and germ-free flies (phases are for cycling metabolites reported in *Figures 1–4*). (**A–D**) Polar histogram plots of the peak phase for oscillating primary metabolites, lipids, and biogenic amines, respectively, in microbiome-containing and sterile flies, using a JTK_cycle value of p<0.05. (**A**) iso31 flies fed on normal chow diets; (**B**) *per01* flies fed on normal chow diets; (**C**) iso31 flies fed on high protein diets; and (**D**) iso31 flies fed on high sugar diets.

(*Figure 1—figure supplement 6A, G and I*). Thus, overall metabolite cycling decreases on high protein and high sugar diets. But while high protein potentiates the effect of the microbiome in promoting cycling, high sugar suppresses it (*Figure 1—figure supplements 2A, D, G, M–R and 6G–J*).

Overall, the results suggest that nutrient-specific diets interact in different ways with the microbiome to regulate the cycling of gut metabolites.

## Circadian clock, dietary composition, and microbiome determine the phase of cycling metabolite in the gut

We further explored how the phase of cycling is affected by the endogenous clock, diet, and microbiome. The phase of cycling in *per01* flies was different from that of wild-type flies. The average phase of primary metabolites in wild-type flies falls around ZT6 while *per01* flies mainly have a peak phase near ZT3. Lipids lose a morning peak in the clock mutant and most dramatically, biogenic amines go from peaking in the middle of the day to the middle of the night (*Figure 5A and B*, left panels). This effect of the host clock on the phase of cycling metabolites was even stronger in microbiome-free flies (*Figure 5A and B*, right panels). Phases were also different on the different diets in microbiome-containing flies, with just primary metabolites conserving a peak around ZT6 in both the standard and high sugar diet; in high protein, these metabolites showed a peak around ZT15. Lipids exhibit a cycling peak primarily around ZT0 and ZT6 in standard diets. However, the phase shifts to around ZT6 and ZT15 for high protein diets, and to around ZT3 and ZT15 for high sugar diets. Biogenic amines have a cycling phase that falls around ZT15 in both high protein and high sugar diets, while in standard diets, the peak occurs around ZT6 (*Figure 5A, C and D* left panels). Germ-free flies also show significant effects of diet composition on cycling phase (*Figure 5*).

## Feeding rhythms have no influence on metabolite cycling

To determine whether metabolite cycling is affected by altered feeding, we investigated food intake at different times of day in different genotypes and under different conditions and diets. Given that much has already been published on timed feeding, and the fact that it does not significantly affect

food consumption (*Gill et al., 2015*; *Villanueva et al., 2019*), we chose to investigate feeding rhythms of AM and AS iso31 and *per01* flies on different diets. The results showed that, compared to normal chow diets, high protein diets decreased food consumption, while high sugar diets increased food consumption (*Figure 4—figure supplement 1A*). Flies fed high protein diets displayed two feeding peaks, one in the morning and one in the evening, as we previously demonstrated with standard diets. The feeding rhythm showed no significant difference between AM and AS flies, in both iso31 and *per01* strains (*Figure 4—figure supplement 1B*). However, high sugar diets reduce the amplitude of feeding rhythms and eliminate the evening feeding peak. Indeed, the feeding rhythm nearly disappears in AS flies, regardless of genotype, on a high sugar diet (*Figure 4—figure supplement 1C*).

## Amino acid metabolism is modulated by the microbiome, timed feeding, and the circadian clock

The extensive metabolic profiling performed here indicates that the gut microbiome is a central piece in generating daily cycles of metabolites and that depending on the host circadian clock and the composition of the diet, the extent and directionality of that modulation varies.

To get a further understanding of the biological implications of this cycling metabolic datasets, we performed pathway enrichment analysis. First, we explored the ad libitum dataset in which we found that absence of a microbiome reduced metabolic rhythms (*Figure 1*). To our surprise, despite the reduction in cycling metabolite number in flies lacking a microbiome, the pathway enrichment identified numerous pathways as significantly enriched in the AS-only cycling group. These included biosynthesis of branched (valine, leucine, and isoleucine) and aromatic (phenylalanine, tyrosine, and tryptophan) amino acids as well as the aminoacyl-tRNA biosynthesis pathway that is relevant for protein synthesis (*Figure 6A*). Another amino acid pathway, involving alanine, aspartate, and glutamate, was found significant in AS and AM cycling sets, with different specific metabolites cycling in these datasets. No pathway related to amino acids or protein metabolism was specific to the AM cycling metabolites, suggesting that, in general, the microbiome abrogates daily oscillations of protein anabolism.

We found that similar pathways were significant in the cycling dataset from microbiome-containing flies subjected to a daytime-restricted feeding paradigm versus ad libitum feeding (*Figure 6B*). Additionally, we determined whether pathways affected by timed feeding were different in flies with and without a microbiome and found that amino acid pathways cycle significantly in both groups (*Figure 6D*). Overall, the data suggest that amino acid and protein biosynthesis cycles are dampened by the gut microbiome but potentiated by time-restricted feeding and the latter has a larger effect in microbiome-containing flies, indicating a microbiome and TF interaction. An interesting point here is how timed feeding influences the same amino acid pathways in microbiome-containing and germ-free flies (e.g., valine, leucine, and isoleucine biosynthesis) but sometimes through distinct cycling metabolites. For instance, threonine, leucine, isoleucine, 4-methyl-2-oxovaleric acid, and valine in TM (*Figure 6—figure supplement 1* and *Supplementary file 1*). In TS, cycling metabolites in this pathway are isoleucine and valine. Likewise, specific metabolites of a pathway showed differential regulation in AS and AM flies. In AS on a high sugar diet, L-threonine and leucine cycle. Another amino acid, 4-aminobutyric acid, cycles in both AM and AS flies (*Figure 6—figure supplement 1* and *Supplementary file 1*). An acetylated derivative of a natural essential amino acid, N-acetyl-l-leucine, uniquely cycles in AS flies but not in AM flies. On the other hand, the TCA cycle appeared as a shared cycling category in all comparisons (AM vs AS, *Figure 6A*; AM vs TM, *Figure 6B*, AM vs per01 AM, *Figure 6E*), suggesting consistency in the rhythmicity of this pathway. Lastly, we aimed to identify pathways whose cycles were exclusive to microbiome-containing flies. We found that alpha-linoleic metabolism cycled significantly in AM flies compared to AS flies (*Figure 6A*). This pathway cycled significantly in AM and TM flies (*Figure 6B*), indicating that its cycling is not dramatically influenced by timed feeding and may be largely dependent on the microbiome. However, in the absence of a microbiome, timed feeding seems to drive cycling of this pathway (see comparison of AS and TS in *Figure 6C*). In addition, cycling of this pathway requires a functional circadian clock as it is lost in *per01*-AM flies. Thus, cycling of alpha-linoleic metabolism depends upon the microbiome and the clock and can also be driven by timed feeding.

Consistent with the results from flies fed a normal chow diet, we observed that germ-free flies fed a high sugar diet showed an enrichment of amino acid-related metabolism in the absence of a microbiome (*Figure 6F*), which indicates that sugar content did not alter the functional category of cycling

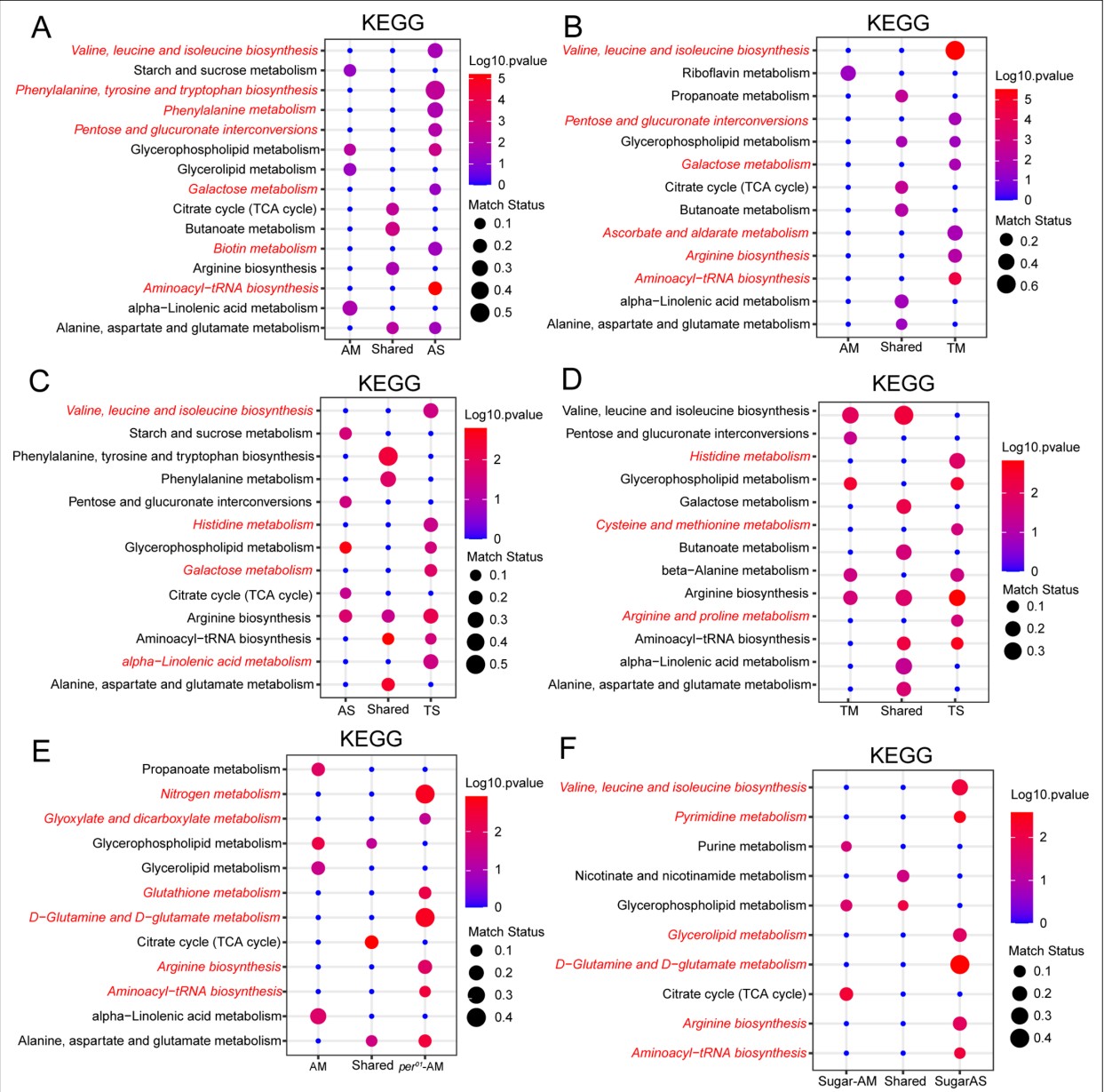

**Figure 6.** Amino acid metabolism is modulated by the microbiome, timed feeding, and circadian clock. (**A**) KEGG enrichment analysis of oscillating metabolites unique to AM flies, shared in AM and AS flies and unique to AS flies. Data only show the p<0.05 results. (**B**) KEGG enrichment analysis of oscillating metabolites unique to AM flies, shared in AM and TM flies and unique to TM flies. Data only show the p<0.05 results. (**C**) KEGG enrichment analysis of oscillating metabolites unique to AS flies, shared in AS and TS flies and unique to TS flies. Data only show the p<0.05 results. (**D**) KEGG enrichment analysis of oscillating metabolites unique to TM flies, shared in TM and TS flies and unique to TS flies. Data only show the p<0.05 results. (**E**) KEGG enrichment analysis of oscillating metabolites unique to *per*[01] AM flies, shared in AM and AS of *per*[01] flies and unique in *per*[01] AS flies. Data only show the p<0.05 results. (**F**) KEGG enrichment analysis of oscillating metabolites unique to high sugar-fed AM flies, shared in AM and AS of *iso31* flies fed on high sugar diets and unique in high sugar-fed AS flies. Data only show the p<0.05 results.

The online version of this article includes the following figure supplement(s) for figure 6:

**Figure supplement 1.** Examples of cycling metabolites in iso31 and *per*[01] flies maintained under different conditions.

metabolites by KEGG analysis. Pathway analysis of cycling metabolites under high protein conditions was not durable due to the reduced number of features.

## Discussion

Our previous study reported that the microbiome dampens transcript cycling in the gut (*Zhang et al., 2023*). By contrast, we now show that the microbiome increases the number of cycling metabolites. At first glance, the two results appear contradictory because enhanced cycling gene expression would appear to be a prerequisite for producing large amounts of metabolites in an oscillatory manner. However, gut microbiomes can effectively provide the host with nutrients and promote metabolite absorption and they may do so in a rhythmic fashion (*Flint, 2012*; *Wang et al., 2017*). Thus, in the presence of a microbiome, the gut may not need to drive cycling of its own genes to promote metabolic oscillations since that might be facilitated by a healthy microbiome. In contrast, in germ-free flies the host might need rhythmic expression of metabolism-related genes to achieve nutritional homeostasis. We report that the cycling metabolites in germ-free flies are extensively enriched in amino acid metabolism. This may compensate for the loss of metabolic properties that would normally be conferred by the microbiome. Interestingly, this pathway is also a major target of timed feeding.

In this study, we first explored the cycling of metabolites in wild flies (iso31) under ad lib and timed feeding conditions, both in germ-free and microbiome-containing flies. To better understand the role of the circadian clock and the effect of different diet compositions on metabolite cycling, we further investigated the microbiome dependence of metabolite oscillations in $per^{01}$ flies fed on normal chow diets and iso31 wild-type flies fed on high protein and high sugar diets, respectively. Although these two sets of studies revealed similar numbers of named metabolites, the total numbers of metabolites exhibited some differences. For instance, the first study had 193/843, 456/4510, and 621/4330 named/total primary metabolites, lipids, and biogenic amines, respectively; in the second study, these numbers were 144/623, 513/2245, and 530/2791, respectively. We acknowledge that due to this fact direct comparison of some of the datasets is not possible. As a strategy to eliminate the influence of different run conditions on the results, we also investigated metabolite cycling profiles using the shared named metabolites across the two batches and found that the results were highly consistent (data not shown). Although not ideal, this suggests that an overall comparison of all the samples included here is appropriate.

Some empirical evidence has found that timed feeding has a beneficial influence on metabolic health (*Gill et al., 2015*; *Lundell et al., 2020*; *Villanueva et al., 2019*). As reported here for metabolite cycling, our previous study found that timed feeding not only enhances gene cycling but also changes the phase of cycling (*Zhang et al., 2023*). Metabolite cycling was further measured under ad lib and timed feeding conditions in microbiome-containing and germ-free flies. In line with the transcript results, timed feeding significantly enhanced metabolite cycling and changed the phase regardless of the presence of a microbiome. However, the microbiome has less effect in timed fed flies, and the reverse was also true, timed feeding had less effect on flies containing a microbiome relative to germ-free flies. It is possible that the microbiome buffers the cyclic metabolome against strong external Zeitgebers such as the time of feeding. Indeed, we previously showed that the gut microbiome stabilizes transcript cycling in the gut such that the phase is not rapidly shifted by acute changes in the light:dark cycle (*Zhang et al., 2023*). Alternatively, given that the presence of a microbiome also has less effect in a timed feeding paradigm, it is possible that TF and the microbiome have overlapping impact on metabolite cycling. On the other hand, oscillations in amino acid metabolism are characteristic of a timed feeding paradigm, as reported here and by others (*Lundell et al., 2020*), and are enhanced in germ-free flies as opposed to microbiome-containing flies. Perhaps these oscillations help the host cope with the lack of microbial driven cycles. How the microbiome helps host cycling remains an open question. We did not detect rhythms of microbial species in *Drosophila*, but it is possible that rhythms of feeding drive rhythms of microbiome abundance in the gut (*Zhang et al., 2023*), given that food is the major source of microbes for flies.

Keeping metabolic rhythms synchronized with the external environment is an important way to maintain optimal fitness for the host. The synchronization is mainly achieved through the circadian clock (*Castanon-Cervantes et al., 2010*; *Dunlap, 1999*; *Roenneberg and Merrow, 2016*), and so mutations in core clock genes can lead to profound metabolic abnormalities (*Marcheva et al., 2010*; *Rudic et al., 2004*). To investigate whether the microbiome-dependent increase

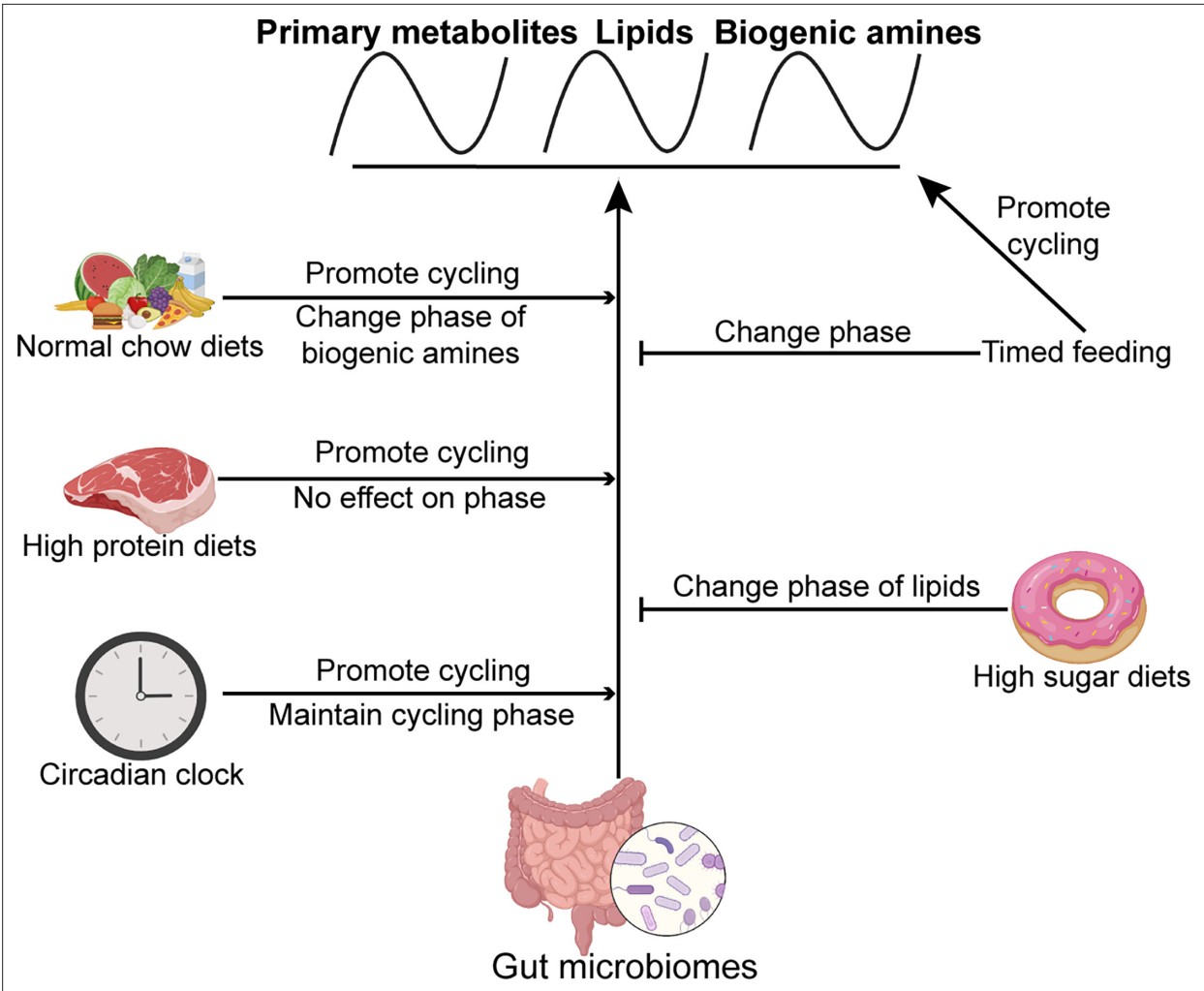

**Figure 7.** Model showing the effects of the microbiome, timed feeding, the circadian clock, and diet composition on metabolite rhythms in the gut. The microbiome broadly increases metabolite rhythms on a normal chow diet and its effects are enhanced by high protein diet and the circadian clock. Timed feeding promotes cycling itself, but attenuates the effect of the microbiome on cycling, as does a high sugar diet. Effects of each of these manipulations on phase are also indicated.

in the number of cycling metabolites depends on the endogenous circadian clock, metabolite cycling was further investigated in microbiome-containing and germ-free $per^{01}$ flies. Interestingly, the number of cycling metabolites was significantly decreased in $per^{01}$ AM flies compared with the $per^{01}$ AS flies. Also, unexpectedly, compared with $per^{01}$ AS flies, cycling metabolites exhibit an opposite cycling phase in $per^{01}$ AM flies. On the other hand, wild-type flies show only a minor change in the phase of cycling primary metabolites with the loss of a microbiome. The cycling phase of biogenic amines shows a significant difference between AM and AS wild-type flies but the difference is larger between AM and AS $per^{01}$ flies. We speculate that the host circadian clock prevents microbial changes from influencing the phase of metabolite cycling. Thus, in clock mutant flies, loss of the microbiome has a dramatic effect on phase. Cycling of metabolites involved in amino acid metabolism occurs in the face of perturbations to the normal, for example, in the absence of a clock ($per^{01}$ AM) or in wild-type flies devoid of a microbiome. This likely reflects compensation for a function normally served by the clock and microbiome (e.g., perhaps the clock drives cycling of a microbiome component) and helps maintain effective amino acid/protein metabolism. Apart from the influence of the endogenous clock, investigation of specific cycling metabolites in the amino acid biosynthesis pathway indicates that although both the microbiome and timed feeding affect this pathway, they modulate different cycling metabolites – and these processes are also

influenced by diet composition. This suggests that the microbiome, timed feeding, circadian clock, and diet can utilize distinct cycling metabolites to fulfill similar metabolism needs. At the same time, the diet may affect the microbiome.

Mice fed on Western diets that contain high sugar amounts, how a significant reduction in the cycling of gut microbiomes, and it has been argued that the changed feeding rhythm may be the reason for this phenomenon (*Frazier et al., 2022*; *Kohsaka et al., 2007*). In the current study, we found that the microbiome suppresses the number of cycling metabolites and substantially affects the phase when flies are fed high sugar diet. Conversely, the microbiome increases the number of cycling metabolites but does not change the cycling phase when flies are fed high protein diets. High sugar diet also resulted in phenotypic differences in the guts of AM and AS flies. In general, on normal chow or high protein diets, AM fly guts had good plasticity and were larger in size than those from AS flies. However, flies fed a high sugar diet had smaller gut sizes than flies fed a high protein or normal chow diets, and the microbiome-mediated gut plasticity difference between AM and AS conditions disappeared (data not shown). Moreover, metabolite abundance analysis showed higher expression of lipids and biogenic amines in AM flies than AS flies when flies were fed on normal chow or high protein diets, but on a high sugar diet AS flies had higher expression of lipids and biogenic amines, although the difference in biogenic amines was not significant (*Figure 1—figure supplement 5*). All these findings suggest that high sugar diets disrupt microbiome function, and this is likely the cause of the increased number of cycling metabolites in AS flies compared to AM flies. While we are attributing differences on high sugar and high protein diets to food composition, we acknowledge that we cannot exclude an effect of altered caloric intake as this was difficult to control for or evaluate.

Changes in the feeding rhythm in response to different diet compositions could contribute to the observed variations in metabolite cycling phenotypes, as seen in mice fed different diets (*Kohsaka et al., 2007*). To address some of these questions, we measured food consumption across 24 hours in flies on different diets, with and without a microbiome. Results indicate that feeding rhythms are not different between AM and AS flies fed on either normal chow or high protein diets (*Zhang et al., 2023*). Given that AM flies show increased metabolite cycling compared to AS flies on normal chow diets, and AS flies show more cycling on high protein diets, these data do not support control of metabolite cycling by feeding rhythms. Furthermore, the feeding rhythm nearly disappears in AS flies fed high sugar diets, regardless of whether they are iso31 or *per01* flies, but AS flies increase metabolite cycling compared to AM flies on high sugar diets. Thus, there seems to be no obvious link between feeding and metabolite rhythms. Caloric content, rather than nutrition (protein/carbohydrate), may be a significant factor affecting metabolite cycling, but this was difficult to control with the diets we used. We note that flies ate more on a high sugar diet and less on a high protein diet, but whether this accounted for loss of the feeding rhythm or was a consequence of it is unclear. Other factors such as locomotor rhythms and sleep are also likely contributors to metabolite cycling and potential subjects of future study. For instance, sleep reduces metabolic rate, which could affect metabolite levels (*Stahl et al., 2017*).

Based on these extensive metabolic profiling data, we propose that an interplay of the microbiome, the circadian clock, and food intake maintain metabolic homeostasis in the gut. With respect to food, the timing of intake as well as the composition are important (*Figure 7*). Interestingly, the microbiome and time-restricted feeding paradigms can compensate for each other to some extent, suggesting that different strategies can be leveraged to serve organismal health.

## Acknowledgements

This work was supported by a grant from the VolkswagenStiftung (Life?). YZ was supported by the Opening Projects of Shanghai Key Laboratory of Chemical Biology and the National Natural Science Foundation of China (31972308). AS is an Investigator of the HHMI.

## Additional information

### Competing interests

Amita Sehgal: Reviewing editor, eLife. The other authors declare that no competing interests exist.

## Funding

| Funder | Grant reference number | Author |
|---|---|---|
| Shanghai Key Laboratory of Chemical Biology | Opening Projects | Yueliang Zhang |
| National Natural Science Foundation of China | 31972308 | Yueliang Zhang |
| VolkswagenStiftung | Life? | Amita Sehgal |

The funders had no role in study design, data collection and interpretation, or the decision to submit the work for publication.

## Author contributions

Yueliang Zhang, Conceptualization, Data curation, Investigation, Methodology, Writing – original draft; Sara B Noya, Data curation, Formal analysis, Writing – review and editing; Yongjun Li, Formal analysis, Methodology, Writing – review and editing; Jichao Fang, Funding acquisition, Project administration; Amita Sehgal, Conceptualization, Supervision, Funding acquisition, Project administration, Writing – review and editing

## Author ORCIDs

Amita Sehgal ⓘ https://orcid.org/0000-0001-7354-9641

Reviewer #1 (Public review): https://doi.org/10.7554/eLife.97130.3.sa1
Reviewer #2 (Public review): https://doi.org/10.7554/eLife.97130.3.sa2
Reviewer #3 (Public review): https://doi.org/10.7554/eLife.97130.3.sa3
Author response https://doi.org/10.7554/eLife.97130.3.sa4

# Additional files

## Supplementary files

Supplementary file 1. JTK_CYCLE analysis was performed on three classes of metabolites (primary metabolites, lipids, and biogenic amines) in the following experimental groups, iso31 flies fed standard diets (AM vs AS, AM vs TM, AS vs TS, TM vs TS), $per^{01}$ flies fed standard diets (AM versus AS), iso31 flies fed high protein and high sugar diets (AM vs AS).

MDAR checklist

## Data availability

Metabolite data have been deposited in MetaboLights under accession codes MTBLS8819.

The following dataset was generated:

| Author(s) | Year | Dataset title | Dataset URL | Database and Identifier |
|---|---|---|---|---|
| Zhang Y, Noya SB, Li Y, Sehgal A | 2025 | The microbiome interacts with the circadian clock and dietary composition to regulate metabolite cycling in the gut | https://www.ebi.ac.uk/metabolights/editor/MTBLS8819/descriptors | MetaboLight, MTBLS8819 |

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
