## [Editor Report · eLife Assessment]

This study presents **valuable** findings about daily rhythm changes of the *Drosophila melanogaster* adult gut metabolome, which is shown to be dependent on the circadian clock genotype, dietary regime and composition, and gut microbiota. The phenomena observed are supported by **convincing** experimental evidence. The general descriptive approach limits the power of the proposed conclusions. The work will be of interest to a broad range of physiology specialists

---

## [Referee Report · Reviewer #1 (Public review)]

The authors build on their previous study that showed the midgut microbiome does not oscillate in *Drosophila*. Here, they focus on metabolites and find that these rhythms are in fact microbiome-dependent. Tests of time-restricted feeding, a clock gene mutant, and diet reveal additional regulatory roles for factors that dictate the timing and rhythmicity of metabolites. The study is well-written and straightforward, adding to a growing body of literature that shows the time of food consumption affects microbial metabolism which in turn could affect the host.

Some additional questions and considerations remain:

(1) The main finding that the microbiome promotes metabolite rhythms is very interesting. Which microbiota are likely to be responsible for these effects? Future work could be done to link specific microbiota linked to some of the metabolic pathways investigated.

(2) TF increases the number of rhythmic metabolites in both microbiome-containing and abiotic flies. This is somewhat surprising given that flies typically eat during the daytime rather than at night, very similar to TF conditions. Future work could be done to restrict feeding to other times of day to see if there is a subsequent shift in the timing of metabolites.

(3) Along these lines, the authors show that Per loss of function reveals a change in the phase of rhythmic metabolites. The authors note that these changes are not due to altered daily feeding rhythms in per mutants. This data suggest Per itself is responsible for these changes. Future work could be done to characterize the mechanisms responsible for these effects.

(4) The calorie content of each diet - normal vs high protein vs high-sugar are different. Future work in this area could consider the possibility of a calorie effect rather than difference in nutrition (protein/carbohydrate) or an effect of high protein/sugar on the microbiome itself.

(5) The supplementary table provided outlining the specific metabolites will be useful for future research in this area.

---

## [Referee Report · Reviewer #2 (Public review)]

The revised version of the paper clarifies the authors' discoveries regarding daily changes in metabolite concentrations in the gut of adult female *Drosophila melanogaster*. The authors have addressed all the questions and made the necessary changes, thereby strengthening the value of the article. They demonstrate that various factors influence metabolite oscillations: circadian clock genotype, dietary regime and composition, and gut microbiota.

The notable strengths of this research article remain unchanged: the originality of the experimental design with multiple conditions tested, the variety of detected metabolites, and the clarity in data presentation.

Among the weaknesses, one may consider the following:

Limitations of potential reproducibility: It is unclear whether another research team would identify the same set of cycling metabolites, although similar conclusions appear robust.

Limitations of generalisation: While the conclusions regarding the influence of microbiota, circadian genotype, and dietary regime may be valid, the specific metabolic pathways affected might differ, whereas specific mechanistic explanations remain elusive.

Accuracy of data interpretation: Addressed in comments to the authors. This point corresponds to interpretations discussed by the authors in the text of the manuscript, including beneficial effects of cycling metabolites and phenomenon of oscillation as a whole, its physiological relevance and lack of proofs for existence of any compensative effects, their relevance to metabolism in the gut.

Nevertheless, the authors have clearly and thoroughly addressed all the reviewers' concerns, enabling a better interpretation of the entire study.

---

## [Referee Report · Reviewer #3 (Public review)]

Summary:

Zhang et al sought to quantify the influence of the gut microbiome on metabolite cycling in a *Drosophila* model with extensive metabolomic profiling in 4 time points over a 24 hour period. The authors report that the microbiome enhances metabolite cycling in a context-dependent manner. The metabolomics data presented are comprehensive and complex, and they open up may new questions. The major strength of the work is the production of a large dataset of metabolites that can be the basis for hypothesis generation for more specific experiments. There are several weaknesses that make some of the conclusions speculative.

Strengths:

The revised manuscript is significantly improved due to the inclusion of new data and expanded analyses, particularly of time-resolved food intake. The dataset is comprehensive and of high value to the community. The experimental design includes multiple metabolomic comparisons across genetic and dietary conditions, specifically, germ-free versus microbially-colonized flies, time-restricted versus ad libitum feeding, high-sugar versus high protein diets, and wildtype genotype versus the per01 clock mutant. Additionally, the cycling of individual metabolites is presented, allowing readers to examine metabolites of interest. The datasets are made publicly available, allowing this resource to benefit the community.

Weaknesses

Many of the statistically significant differences, e.g. the effects of the microbiome on lipids and biogenic amines in Fig S5A, are quite small in magnitude, and, thus, it is difficult to believe that they are of biological significance without more mechanistic studies. Key conclusions, such as those pertaining to regulation or compensation by the microbiome, are not fully supported by mechanistic experiments. The manuscript uses terms like "regulate" or "compensate," which imply causality or a purpose of the microbiome that is not yet demonstrated, but this type of study opens up many important questions for which new hypotheses can be formed.

A minor limitation is the modest temporal resolution (only four time points in 24 hours), which constrains interpretation of rhythmicity and phase. Additional experimental controls and targeted perturbation experiments are needed to support conclusions about functional impacts of metabolite oscillations. However, these types of limitations are expected from an early study in the field such as this one. Overall, the data are valuable, and the findings demonstrate the promise of the model for studying the interplay between the microbiome, metabolome, and circadian rhythm.

Assessment of Aims

The authors explore how the microbiome interacts with host circadian rhythms and diet to shape metabolite cycling. They largely succeed in characterizing broad trends and generating a valuable resource dataset. However, the conclusion that the microbiome actively regulates or compensates for cycling under specific conditions is not convincingly demonstrated with the current data.

Impact and Utility

The dataset will be a useful reference for researchers interested in microbiome-host interactions, metabolomics, and circadian biology. Its primary value lies in descriptive insight rather than mechanistic resolution. An alternative perspective is that per01 mutants serve as a useful negative control for rhythmicity detection, providing a baseline for distinguishing signal from experimental noise ---an idea that could be emphasized more in the interpretation.

Contextual Considerations

Metabolomics datasets are valuable for understanding the influence of the microbiome. Future follow-up work using higher resolution sampling and functional perturbations (e.g., more extensive genetic or microbial manipulations) will be essential to test hypotheses about the roles of specific metabolites, regulatory pathways, and microbiota members in circadian modulation. This paper lays a strong foundation for such studies.

---

## [Author Response]

The following is the authors’ response to the original reviews

**Public Reviews:**

**Reviewer #1 (Public Review):**
The authors build on their previous study that showed the midgut microbiome does not oscillate in *Drosophila*. Here, they focus on metabolites and find that these rhythms are in fact microbiome-dependent. Tests of time-restricted feeding, a clock gene mutant, and diet reveal additional regulatory roles for factors that dictate the timing and rhythmicity of metabolites. The study is well-written and straightforward, adding to a growing body of literature that shows the time of food consumption affects microbial metabolism which in turn could affect the host.

We thank the reviewer for the positive comments.

Some additional questions and considerations remain:(1) The main finding that the microbiome promotes metabolite rhythms is very interesting. Which microbiota are likely to be responsible for these effects? The author's previous work in this area may shed light on this question. Are specific microbiota linked to some of the metabolic pathways investigated in Figure 5?

This is a good question. Although the *Drosophila* microbiome shows limited diversity, comprised largely of two major families (*Acetobacteraceae* and *Lactobacillaceae*), effects on the host could arise from just a subset of species within these families. However, identifying these would require inoculating microbiome-free flies with single and mixed combinations of species and conducting metabolomics to examine cycling of each of the three categories of metabolites we studied-- primary, lipids and biogenic amines (each of these may respond differently to different species). We believe this is beyond the scope of this manuscript, which is focused on how cycles in these different types of metabolites change in the context of the microbiome, the circadian clock and different diets.

(2) TF increases the number of rhythmic metabolites in both microbiome-containing and abiotic flies in Figure 1. This is somewhat surprising given that flies typically eat during the daytime rather than at night, very similar to TF conditions. I would have assumed that in a clock-functioning animal, the effect of restricting food availability should not make a huge difference in the time of food consumption, and thus downstream impacts on metabolism and microbiome. Can the authors measure food intake directly to compare the ad-lib vs TF flies to see if there are changes in food intake? Would restricting feeding to other times of day shift the timing of metabolites accordingly?

Previous studies have indicated that there is no significant difference in food consumption between ad-lib and TF flies (Gill et al., 2015; Villaneuva et al 2019). We also found that the presence of a microbiome does not alter total food consumption when compared with germ-free flies (Zhang et al, 2023, and current manuscript). Though flies primarily feed during the day, some food consumption occurs at night (i.e the feeding rhythm is not tight) and so restricting food to the daytime can increase metabolite cycling. Restricting feeding to other times of day is expected to shift metabolite cycling. We previously showed that this shifts transcript cycling (Xu et al, Cell Metabolism 2011)

(3) In Figure 2, Per loss of function reveals a change in the phase of rhythmic metabolites. In addition, the effect of the microbiome on these is very different = The per mutants show increased numbers of rhythmic metabolites when the microbiome is absent, unlike the controls. Is it possible that these changes are due to altered daily feeding rhythms in per mutants? Testing the time and amount of food consumed by the per mutant flies would address this question. Would TF in the per mutants rescue their metabolite rhythms and make them resemble clock-functioning controls?

We previously showed that *per01* flies lose feeding rhythms in DD and LD conditions, but consume a lot more food (Barber et al, 2021). Given that locomotor rhythms are maintained in *per01* in LD (Konopka and Benzer 1971), these rhythms or other rhythms driven by LD cues likely account for the maintenance of metabolite rhythms. And the increased food consumption may contribute to the changes observed. To address the reviewer’s question about the microbiome, we assayed feeding rhythms in *per01* in the absence/presence of a microbiome on the diets that haven’t been tested before (high sugar and high protein diet). Surprisingly, feeding was rhythmic on a high protein diet, regardless of whether a microbiome was present (Figure 4—figure supplement 1). On a high sugar diet, feeding appears to be somewhat rhythmic in the presence of a microbiome (although not significant) and not when the microbiome is absent. The same is true in iso31 controls, and in all cases, the phase is the same. Despite the similar effect of the microbiome on feeding rhythms in wild type and *per01*, the effect on cycling is very different. Thus, feeding rhythms do not appear to explain the effects of the microbiome on metabolite cycling in *per01*.

(4) The calorie content of each diet-normal vs high protein vs high-sugar are different. The possibility of a calorie effect rather than a difference in nutrition (protein/carbohydrate) should be discussed. Another issue worth considering is the effect of high protein/sugar on the microbiome itself. While the microbiome doesn't seem to affect rhythms in the high-protein diet, the high-sugar diet seems highly microbiome-dependent in Supplementary Fig 8C vs D. Does the diet impact the microbiome and thus metabolite rhythmicity downstream?

Thank you for these good suggestions. We have added to the discussion the possibility that caloric content, rather than nutrition (protein/carbohydrate), affects metabolite cycling in flies fed normal vs. high-protein vs. high-sugar diets. We have also discussed the possibility that effects of different diets on metabolite cycling are mediated by changes in the microbiome. We cite papers that show effects of diet on microbiomes.

(5) It would be good if a supplementary table was provided outlining the specific metabolites that are shown in the radial plots. It is not clear if the rhythms shown in the figures refer to the same metabolites peaking at the same time, or rather the overall abundance of completely different metabolites. This information would be useful for future research in this area.

We have added a supplementary Supplementary file 1 which includes all the raw metabolites.

**Reviewer #2 (Public Review):**
Summary:The paper addresses several factors that influence daily changes in concentration of metabolites in the *Drosophila melanogaster* gut. The authors describe metabolomes extracted from fly guts at four time-points during a 24-hour period, comparing profiles of primary metabolites, lipids, and biogenic amines. The study elucidates that the percentage of metabolites that exhibit a circadian cycle, peak phases of their increased appearance, and the cycling amplitude depends on the combination of factors (microbiome status, composition or timing of the diet, circadian clock genotype). Multiple general conclusions based on the data obtained with modern metabolomics techniques are provided in each part of the article. Descriptive analysis of the data supports the finding that microbiome increases the number of metabolites for which concentration oscillates during the day period. Results of the experiments show that timed feeding significantly enhanced metabolite cycling and changed its phase regardless of the presence of a microbiome. The authors suggest that the host circadian rhythm modifies both metabolite cycling period and the number of such metabolites.Strengths:The obvious strength of the study is the data on circadian cycling of the detected 843, 4510, and 4330 total primary metabolites, lipids, and biogenic amines respectively in iso31 flies and 623, 2245, and 2791 respective metabolites in *per01* mutants. The comparison of the abundance of these metabolites, their cycling phase, and the ratio of cycling/non-cycling metabolites is well described and illustrated. The conditions tested represent significant experimental interest for contemporary chronobiology: with/without microbiota, wild-type/mutant period gene, ad libitum/time-restricted feeding, and high-sugar/high-protein diet. The authors conclude that the complex interaction between these factors exists, and some metabolic implications of combinations of these factors can be perceived as reminiscent of metabolic implications of another combination ("...the microbiome and time-restricted feeding paradigms can compensate for each other, suggesting that different strategies can be leveraged to serve organismal health"). Enrichment analysis of cycling metabolites leads to an interesting suggestion that oscillation of metabolites related to amino acids is promoted by the absence of microbiota, alteration of circadian clock, and time-restricted feeding. In contrast, association with microbiota induces oscillation of alpha-linolenic acid-related metabolites. These results provide the initial step for hypothesising about functional explanations of the uncovered observations.

We thank the reviewer for summarizing the contributions made by this manuscript.

Weaknesses:Among the weaknesses of the study, one might point out too generalist interpretations of the results, which propose hypothetical conclusions without their mechanistic proof. The quantitative indices analysed are obviously of particular interest, however are not self-explaining and exhaustive. More specific biological examples would add valuable insights into the results of this study, making conclusions clearer. More specific comments on the weaknesses are listed below:(1) The criterion of the percentage of cycling metabolites used for comparisons has its own limitations. It is not clear, whether the cycling metabolites are the same in the guts with/without microbiota, or whether there are totally different groups of metabolites that cycle in each condition. GO enrichment analysis gives only a partial assessment, but is still not quantitative enough.

Microbiome-containing flies and germ-free flies do share some cycling metabolites. Figure 6 provides GO analysis for the pathways enriched in each condition, and Figure 1—figure supplement 2 shows quantitative data on the number that overlap between different conditions. We have also expanded discussion of specific cycling groups (e.g. amino acid metabolism) to indicate that different metabolites of the same pathway may cycle under different conditions. In addition, we have added detailed information for all cycling metabolites in Supplementary file 1.

(2) The period of cycling data is based on only 4 time points during 24 hours in 4 replicates (>200 guts per replicate) on the fifth day of the experiment (10-12-day-old adults). It does not convincingly prove that these metabolites cycle the following days or more finely within the day. Moreover, it is not clear how peaks in polar histogram plots were detected in between the timepoints of ZT0, ZT6, ZT12, and ZT18.

We acknowledge these limitations, but note that these experiments are very challenging because of the amount of tissue/guts needed for each data point and the time it takes to dissect each gut. Thus, getting more closely spaced time points is difficult. And we believe the detection of daily peaks with four biological replicates provides good evidence for cycling. The peaks in polar histogram plots are based on the parameter of JTK_adjphase when conducting JTK cycle analysis; as the data are averaged across replicates, the average can sometimes fall in between two assayed time points. Details can be found in the attached Supplementary Tables.

(3) Average expression and amplitude are analysed for groups of many metabolites, whereas the data on distinct metabolites is hidden behind these general comparisons. This kind of loss of information can be misleading, making interpretation of the mentioned parameters quite complicated for authors and their readers. Probably more particular datasets can be extracted to be discussed more thoroughly, rather than those general descriptions.

We analyzed groups of metabolites, dividing them into primary metabolites, lipids and biogenic amines, to extract general take-home messages and also to simplify the presentation. Figure 6 demonstrates specific pathways whose cycling is affected in each condition assayed. And Figure S11 shows examples of cycling metabolites under different conditions. To highlight a dataset that is altered under different conditions, we expanded our discussion of amino acid metabolism, and show how the specific metabolites that cycle in this pathway may vary from one condition to another (Figure 6—figure supplement 1). For more quantitative data on individual metabolites, we now provide supplementary tables that display all the cycling metabolites. These include those uniquely cycling in one group, those shared between both two groups, and those uniquely cycling in the other group.

(4) The metabolites' preservation is crucial for this type of analysis, and both proper sampling plus normalisation require more attention. More details about measures taken to avoid different degradation rates, different sizes of intestines, and different amounts of microbes inside them will be beneficial for data interpretation.

We were careful to control for gut size and to preserve the samples so as to minimize degradation (We placed all the fly samples on ice during collection, and the entire dissection process was also conducted on ice. Once the gut sample collection was completed, we immediately transferred the samples to dry ice for storage. After we finished collecting all the samples, we stored them at -80°C). In general, gut sizes varied in the following order: females fed high-protein diets >females fed normal chow diets> female flies fed high-sugar diets. As the metabolomic facility suggested 10mg samples for each biological repeat, we dissected at least 180 female guts from flies fed high-protein diets, 200 female guts from flies fed normal chow diets, and at least 250 female guts from flies fed high-sugar diets. Also, as gut sizes were smaller in sterile flies, relative to microbiome-containing flies, on a high protein diet, we collected 200 guts from sterile flies under these conditions. Finally, the service that conducted the metabolomics (UC Davis) provided three detailed files to describe the extraction process for primary metabolites, lipids, and biogenic amines, respectively. We have submitted these files as supplemental materials in the revised manuscript.

(5) The data in the article describes formal phenomena, not directly connected with organism physiology. The parameters discussed obviously depend on the behavior of flies. Food consumption, sleep, and locomotor activity could be additionally taken into account.

These are very interesting suggestions. Previous results indicated that microbiome-containing flies do not change their total food consumption or exhibit changes in feeding rhythms when compared with germ-free flies (Zhang et al., 2023), which indicates that microbiome-mediated metabolite cycling is independent of feeding rhythms. As noted above, we examined the contribution of feeding to metabolite cycling in *per01* flies, and did not see an obvious link. We also assayed feeding rhythms and overall food consumption in wild type under AS and AM conditions and on different diets, and likewise could not account for changes in metabolite cycling based on altered food intake (Figure 4—supplement 1). We acknowledge that behavior, including locomotor activity and sleep, could indeed influence metabolite cycling. We have added discussion of this.

(6) Division of metabolites into three classes limits functional discussion of found differences. Since the enrichment analysis provided insights into groups of metabolites of particular interest (for example, amino acid metabolism), a comparison of their cycling characteristics can be shown separately and discussed.

The intent of this work was to provide an overall account of changes in metabolite cycling that occur under different conditions/diets/genotypes. We have expanded the discussion of amino acid metabolism and show how different metabolites of this pathway cycle under different conditions (Figure 6—figure supplement 1). We believe that discussion/analysis of other specific groups would be good follow-up studies, which can build upon this work. Detailed datasets about all cycling metabolites are provided in Supplementary file 1.

**Reviewer #3 (Public Review):**
Summary:The authors. sought to quantify the influence of the gut microbiome on metabolite cycling in a *Drosophila* model with extensive metabolomic profiling over a 24-hour period. The major strength of the work is the production of a large dataset of metabolites that can be the basis for hypothesis generation for more specific experiments. There are several weaknesses that make the conclusions difficult to evaluate. Additional experiments to quantify food intake over time will be required to determine the direct role of the microbiome in metabolite cycling.

Strengths:

An extensive metabolomic dataset was collected with treatments designed to determine the influence of the gut microbiome on metabolite circadian cycling.

Weaknesses:(1) The major strength of this study is the extensive metabolomic data, but as far as I can tell, the raw data is not made publicly available to the community. The presentation of highly processed data in the figures further underscores the need to provide the raw data (see comment 3).

The raw data have been submitted to the public metabolite database. https://www.ebi.ac.uk/metabolights/. (ID: MTBLS8819)

In addition, the normalized metabolite data have been added in the supplemental materials.

(2) Feeding times heavily influence the metabolome. The authors use timed feeding to constrain when flies can eat, but there is no measurement of how much they ate and when. That needs to be addressed.Since food is the major source of metabolites, the timing of feeding needs to be measured for each of the treatment groups. In the previous paper (Zhang et al 2023 PNAS), the feeding activity of groups of 4 male flies was measured for the wildtype flies. That is not sufficient to determine to what extent feeding controls the metabolic profile of the flies. Additionally, timed feeding opportunities do not equate to the precise time of feeding. They may also result in dietary restriction, leading to the loss of stress resistance in the TF flies. The authors need to measure food consumption over time in the exact conditions under which transcriptomic and metabolomic cycling are measured. I suggest using the EX-Q assay as it is much less effort than the CAFE assay and can be more easily adapted to the rearing conditions of the experiments.

As noted above, we have now added considerable additional data on feeding and feeding rhythms in microbiome-containing and sterile wild type and *per01* flies on different diets (Figure 4—figure supplement 1). Our previous study, using the EX-Q assay method, found no differences in either total food consumption or feeding rhythms between microbiome-containing flies and germ-free flies (Zhang et al., 2023). Also, previous work has demonstrated that there is no significant difference in food consumption between ad-lib and TF flies (Villaneuva et al 2019).

(3) The data on the cycling of metabolites is presented in a heavily analyzed form, making it difficult to evaluate the validity of the findings, particularly when a lack of cycling is detected. The normalization to calculate the change in cycling due to particular treatments is particularly unclear and makes me question whether it is affecting the conclusions. More presentation of the raw data to show when cycling is occurring versus not would help address this concern, as would a more thorough explanation of how the normalization is calculated - the brief description in the methods is not sufficient.For instance, the authors state that "timed feeding had less effect on flies containing a microbiome relative to germ-free flies." One alternative interpretation of that result is that both treatments are cycling but that the normalization of one treatment to the other removes the apparent effect. This concern should be straightforward to address by showing the raw data for individual metabolites rather than the group.

We have added Supplement Table1-21 that includes detailed information on metabolite identity and data processing. Also, we have included the raw data, encompassing all the cycling metabolites, in the Supplementary file 1.

**Recommendations for the authors:**

**Reviewer #1 (Recommendations For The Authors):**
(1) The abstract could be rewritten to clarify. I found the last part of the introduction better but struggled to understand the abstract.

We apologize for this. The abstract was indeed quite dense; we have revised it for clarity.

(2) Supplementary Figure 8 could be moved to the main text. Since all the comparisons are on one page it is much easier to see the similarities and differences in the conditions tested.

We have moved Supplementary Figure 8 to main Figure 5.

(3) The sex and age of the flies used in all experiments should be clarified. The authors mention female guts are collected in the methods (line 111) but it is not clear if this is throughout.

All guts used in this study were female. We have clarified this in the manuscript.

**Reviewer #2 (Recommendations For The Authors):**
Some minor notes that might be improved:(1) The order of obtaining eggs without microbiota might be different (first - bleaching, second - sterilisation with ethanol). Otherwise, it is not clear why dechorionating is needed after sterilisation.

Protocols for generating axenic flies vary. We used the method Feltzin et al reported in 2019: “For newborn fly embryos (<12 hours). First, cleanse and sterilize any leftover agar from collection plates using 100% ethanol, second, dechorionate the fly embryos with 10% bleach, and then immediately rinse three times in germ-free PBS”.

(2) References for the resources used might be provided (MetaboAnalyst5.0, JTK_CYCLEv3.1).

We have added the reference for MetaboAnalyst5.0, JTK_CYCLEv3.1 (Pang et al., 2022)

(3) References or justification for the chosen composition of the diets might be useful (standard diet, high-protein diet, high-sugar diet).

We have added the references (Bedont et al, 2021, Morris et al, 2021).

(4) Justification of the choice of iso31 line and *per01* mutant might be important.

iso31 is the standard wild type line we use in the laboratory. To understand the role of the endogenous clock in microbiome-mediated metabolite cycling, we chose the classical canonical clock mutant *per01* as this displays fewer non-circadian phenotypes seen. For instance, loss of transcriptional activators of the clock produces additional effects (e.g. hyperactivity), likely because of the effect it has on overall expression of many genes. We have added this explanation to the manuscript.

(5) Abbreviation decoding might be introduced when it is used for the first time in the text (line 240 - TM, TS).

We apologize for this omission and have rectified it. Thanks

TM (timed feeding microbiome-containing flies)

TS (timed feeding germ-free flies)

(6) The term "germ-free" is recommended to be avoided in the context of the paper (germ-free = infertile for animals). It might be replaced with the terms "without microbiota" or "germ-free" for example.

Given that the reviewer recommends use of the word “germ-free” in the second sentence, we assume that the first sentence was intended to say we should avoid “sterile” (and not “germ-free”). We have edited to “germ-free” in the manuscript.

(7) When only one diet is assumed, it might be better to say so (line 324 - "the protein diet" instead of "protein diets").

Sorry, we have edited accordingly.

(8) Too many speculative conclusions are confusing (line 476 - what does it mean for "just as” - how exactly similar; line 477 - what kind of "compensation"; line 503 - how exactly it is related to "metabolic homeostasis" and to which kind of homeostasis).

“just as” was not intended to refer to any degree of similarity but only to the fact that amino acid cycling occurs in the absence of a clock, as it does in the absence of a microbiome. We speculate that this “compensates” for something that is normally conferred by the clock and the microbiome, for instance maybe the clock drives cycling of a microbiome component that is important for protein metabolism. In the absence of either the clock or the microbiome, this is compensated for by amino acid cycling. We have clarified in the text.

We used the term "metabolic homeostasis" to reflect steady maintenance of metabolic health via interaction and modulation of different factors. As in the case of the example given above for amino acid metabolism, a perturbation of one process might produce a change in another to optimize metabolism. We have changed the wording in the text to better convey our message (lines 576-579)

(9) Particular examples of metabolites might be beneficial for supporting conclusions (a figure which shows, for instance, the specific data on linolenic acid: in which conditions it cycles, in which not, what is the period of cycling, what are the exact expression and JTK_amplitude values).

All cycling metabolites, including linolenic acid, are now included in the supplemental tables.

**Reviewer #3 (Recommendations For The Authors):**
(1) The level of biological replication is unclear for the metabolomic experiments. I see that 200 guts per sample were collected and 4 repeat samples were made at each timepoint. Are these 4 biological replicates for each treatment (AS, AM, TS, TM) at each timepoint? 5 replicates are standard in metabolomics. Please be more explicit in the methods.

There are 4 biological replicates for each time point of each of the 4 treatments. The metabolomics service recommended 4-6 replicates, so we prepared 4 replicates for each sample. As noted above, these preparations are quite difficult. We found that in general the biological replicates do not differ significantly from each other.

(2) Wolbachia can have a significant influence on fly physiology. How was this variable addressed? Were flies checked for Wolbachia?

All the flies are *Wolbachia*-free, as in our previous study (Zhang et al., 2023). Initially, we treated the flies with 1 mM kanamycin (11815024, ThermoFisher) to remove bacteria. Afterwards, we repopulated the flies with a *Wolbachia*-free microbiome containing *Lactobacillus* and *Acetobacter* bacteria from a medium previously occupied by other flies.

(3) In Results section 1, the authors report changes in the percentages of metabolites that are cycling, but no statistical test is presented to show that these changes are indeed significant. The authors need to report statistics on the percentages of cycling metabolites.

We used statistical tests, specifically JTK cycle, to determine cycling of each metabolite. The P value for cycling of each metabolite in this test is computed on the basis of all the biological replicates and all time points. Metabolites that showed a significant P value contribute to the percent cycling. As a result, there is only one value for the percentage cycling in each condition. Thus, statistical analysis cannot be done.

(4) The authors report that the species proportions in the gut microbiome don't cycle, but do absolute CFU counts? By many accounts (see e.g. Blum et al 2013 mBio), the gut microbiome in lab flies is what they recently ate from the food. The abundance of bacteria in the gut would then be directly tied to the timing of feeding. Timed feeding should produce oscillations in individual flies, so individual flies should be analyzed.

We assume the reviewer is suggesting that rhythmic feeding could result in rhythmic abundance of the microbiome, which could contribute to cycling. This is indeed a possibility and one we now discuss in the manuscript. Thanks! Analysis of the gut microbiome in individual flies would require quantitation of CFUs from single guts. We do not believe a single gut would yield enough material.

(5) Line 252: the ZT9 peak could just be due to feeding and digestion.

This is possible. We now acknowledge this

(6) What is the expectation for metabolite cycling in per mutant flies? Shouldn't per mutant flies have no cycling on average? Does the cycling suggest there is an external factor causing cycling?

Under light-dark conditions, metabolite cycling in *per* mutant flies may be driven by light: dark cues, either directly or through other light-driven rhythms e,g. locomotor activity is rhythmic in *per01* flies maintained in LD.

(7) Performing food intake analysis on each of the treatments would provide critical data to address the direct role of the microbiome in metabolite cycling.

As noted above, we now provide considerable additional data on food intake at different times of day in microbiome-containing and germ-free wild type and *per01* flies on different diets (Figure 4—figure supplement 1). Overall, our data indicate that food intake or feeding rhythms do not account for the effects we report here.

(8) Please be more explicit about replication in the methods and figure legends.

We have added n=4 for each condition in the methods and figure legends.

(9) There are numerous minor grammatical errors such as incorrect verb tenses and usage of articles. Additional proofreading could correct these.

Sorry! We have done a thorough proofreading and made corrections.